# Optimal Transport-Guided Conditional Score-Based Diffusion Model

**Xiang Gu[1], Liwei Yang[1], Jian Sun (✉)[1,2,3], Zongben Xu[1,2,3]**
[1] School of Mathematics and Statistics, Xi'an Jiaotong University, Xi'an, China
[2] Pazhou Laboratory (Huangpu), Guangzhou, China
[3] Peng Cheng Laboratory, Shenzhen, China
`{xianggu,yangliwei}@stu.xjtu.edu.cn {jiansun,zbxu}@xjtu.edu.cn`

## Abstract

Conditional score-based diffusion model (SBDM) is for conditional generation of target data with paired data as condition, and has achieved great success in image translation. However, it requires the paired data as condition, and there would be insufficient paired data provided in real-world applications. To tackle the applications with partially paired or even unpaired dataset, we propose a novel Optimal Transport-guided Conditional Score-based diffusion model (OTCS) in this paper. We build the coupling relationship for the unpaired or partially paired dataset based on $L_2$-regularized unsupervised or semi-supervised optimal transport, respectively. Based on the coupling relationship, we develop the objective for training the conditional score-based model for unpaired or partially paired settings, which is based on a reformulation and generalization of the conditional SBDM for paired setting. With the estimated coupling relationship, we effectively train the conditional score-based model by designing a "resampling-by-compatibility" strategy to choose the sampled data with high compatibility as guidance. Extensive experiments on unpaired super-resolution and semi-paired image-to-image translation demonstrated the effectiveness of the proposed OTCS model. From the viewpoint of optimal transport, OTCS provides an approach to transport data across distributions, which is a challenge for OT on large-scale datasets. We theoretically prove that OTCS realizes the data transport in OT with a theoretical bound. Code is available at `https://github.com/XJTU-XGU/OTCS`.

## 1   Introduction

Score-based diffusion models (SBDMs) [1–8] have gained much attention in data generation. SBDMs perturb target data to a Gaussian noise by a diffusion process and learn the reverse process to transform the noise back to the target data. The conditional SBDMs [2, 9–13] that are conditioned on class labels, text, low-resolution images, *etc.*, have shown great success in image generation and translation. The condition data and target data in the conditional SBDMs [2, 9–13] are often paired. That is, we are given a condition for each target sample in training, *e.g.*, in the super-resolution [10, 14, 15], each high-resolution image (target data) in training is paired with its corresponding low-resolution image (condition). However, in real-world applications, there could not be sufficient paired training data, due to the labeling burden. Therefore, it is important and valuable to develop SBDMs for applications with only unpaired or partially paired training data, *e.g.*, unpaired [16] or semi-paired [17] image-to-image translation (I2I). Though there are several SBDM-based approaches [18–22] for unpaired I2I, the score-based models in these approaches are often unconditioned, and the conditions are imposed in inference by cycle consistency [21], designing the initial states [19, 22], or adding a guidance term to the output of the unconditional score-based model [18, 20]. It is unclear how to train the conditional score-based model with unpaired training dataset. For the task with a few paired and a large number

of unpaired data, *i.e.*, partially paired dataset, there are few SBDMs for tackling this task, to the best of our knowledge.

This paper works on how to train the conditional score-based model with unpaired or partially paired training dataset. We consider the I2I applications in this paper where the condition data and target data are images respectively from different domains. The main challenges for this task are: 1) the lack of the coupling relationship between condition data and target data hinders the training of the conditional score-based model, and 2) it is unclear how to train the conditional score-based model even with an estimated coupling relationship, because it may not explicitly provide condition-target data pairs as in the setting with paired data. We propose a novel Optimal Transport-guided Conditional Score-based diffusion model (OTCS) to address these challenges. Note that different from the existing OT-related SBDMs that aim to understand [23] or promote [24, 25] the unconditional score-based models, our approach aims to develop the conditional score-based model for unpaired or partially paired data settings guided by OT.

We tackle the first challenge based on optimal transport (OT). Specifically, for applications with unpaired setting, *e.g.*, unpaired super-resolution, practitioners often attempt to translate the condition data (*e.g.*, low-resolution images) to target domain (*e.g.*, consisting of high-resolution images) while preserving image structures [26], *etc*. We handle this task using unsupervised OT [27] that transports data points across distributions with the minimum transport cost. The coupling relationship of condition data and target data is modeled in the transport plan of unsupervised OT. For applications with partially paired setting, it is reasonable to utilize the paired data to guide building the coupling relationship of unpaired data, because the paired data annotated by humans should have a reliable coupling relationship. The semi-supervised OT [28] is dedicated to leveraging the annotated keypoint pairs to guide the matching of the other data points. So we build the coupling relationship for partially paired dataset using semi-supervised OT by taking the paired data as keypoints.

To tackle the second challenge, we first provide a reformulation of the conditional SBDM for paired setting, in which the coupling relationship of paired data is explicitly considered. Meanwhile, the coupling relationship in this reformulation is closely related to the formulation of the coupling from $L_2$-regularized OT. This enables us to generalize the objective of the conditional SBDM for paired setting to unpaired and partially paired settings based on OT. To train the conditional score-based model using mini-batch data, directly applying the standard training algorithms to our approach can lead to sub-optimal performance of the trained conditional score-based model. To handle this challenge, we propose a new "resampling-by-compatibility" strategy to choose sampled data with high compatibility as guidance in training, which shows effectiveness in experiments.

We conduct extensive experiments on unpaired super-resolution and semi-paired I2I tasks, showing the effectiveness of the proposed OTCS for both applications with unpaired and partially paired settings. From the viewpoint of OT, the proposed OTCS offers an approach to transport the data points across distributions, which is known as a challenging problem in OT on large-scale datasets. The data transport in our approach is realized by generating target samples from the optimal conditional transport plan given a source sample, leveraging the capability of SBDMs for data generation. Theoretically and empirically, we show that OTCS can generate samples from the optimal conditional transport plan of the $L_2$-regularized unsupervised or semi-supervised OTs.

## 2 Background

Our method is closely related to OT and conditional SBDMs, which will be introduced below.

### 2.1 Conditional SBDMs with Paired Data

The conditional SBDMs [2, 6, 9–13] aim to generate a target sample $\mathbf{y}$ from the distribution $q$ of target training data given a condition data $\mathbf{x}$. For the paired setting, each target training sample $\mathbf{y}$ is paired with a condition data $\mathbf{x}_{\text{cond}}(\mathbf{y})$. The conditional SBDMs with paired dataset can be roughly categorized into two types, respectively under the classifier guidance [6] and classifier-free guidance [2, 9–13]. Our approach is mainly related to the second type of methods. These methods use a forward stochastic differential equation (SDE) to add Gaussian noises to the target training data for training the conditional score-based model. The forward SDE is $\mathrm{d}\mathbf{y}_t = f(\mathbf{y}_t, t)\,\mathrm{d}t + g(t)\,\mathrm{d}\mathbf{w}$ with $\mathbf{y}_0 \sim q$, and $t \in [0, T]$, where $\mathbf{w} \in \mathbb{R}^D$ is a standard Wiener process, $f(\cdot, t) : \mathbb{R}^D \to$

$\mathbb{R}^D$ is the drift coefficient, and $g(t) \in \mathbb{R}$ is the diffusion coefficient. Let $p_{t|0}$ be the conditional distribution of $\mathbf{y}_t$ given the initial state $\mathbf{y}_0$, and $p_t$ be the marginal distribution of $\mathbf{y}_t$. We can choose the $f(\mathbf{y}, t)$, $g(t)$, and $T$ such that $\mathbf{y}_t$ approaches some analytically tractable prior distribution $p_{\text{prior}}(\mathbf{y}_T)$ at time $t = T$, *i.e.*, $p_T(\mathbf{y}_T) \approx p_{\text{prior}}(\mathbf{y}_T)$. We take $f, g, T$, and $p_{\text{prior}}$ from two popular SDEs, *i.e.*, VE-SDE and VP-SDE [3] (please refer to Appendix A for model details). The conditional score-based model is trained by denoising score-matching loss:

$$\mathcal{J}_{\text{DSM}}(\theta) = \mathbb{E}_t w_t \mathbb{E}_{\mathbf{y}_0 \sim q} \mathbb{E}_{\mathbf{y}_t \sim p_{t|0}(\mathbf{y}_t|\mathbf{y}_0)} \left\| s_\theta(\mathbf{y}_t; \mathbf{x}_{\text{cond}}(\mathbf{y}_0), t) - \nabla_{\mathbf{y}_t} \log p_{t|0}(\mathbf{y}_t|\mathbf{y}_0) \right\|_2^2, \quad (1)$$

where $w_t$ is the weight for time $t$. In this paper, $t$ is uniformly sampled from $[0, T]$, *i.e.*, $t \sim \mathcal{U}([0, T])$. With the trained $s_{\hat{\theta}}(\mathbf{y}; \mathbf{x}, t)$, given a condition data $\mathbf{x}$, the target sample $\mathbf{y}_0$ is generated by the reverse SDE as $d\mathbf{y}_t = \left[ f(\mathbf{y}_t, t) - g(t)^2 s_{\hat{\theta}}(\mathbf{y}_t; \mathbf{x}, t) \right] dt + g(t) d\bar{\mathbf{w}}$, where $\bar{\mathbf{w}}$ is a standard Wiener process in the reverse-time direction. This process starts from a noise sample $\mathbf{y}_T$ and ends at $t = 0$.

## 2.2 Optimal Transport

**Unsupervised OT.** We consider a source distribution $p$ and a target distribution $q$. The unsupervised OT [29] aims to find the optimal coupling/transport plan $\pi$, *i.e.*, a joint distribution of $p$ and $q$, such that the transport cost is minimized, formulated as the following optimization problem:

$$\min_{\pi \in \Gamma} \mathbb{E}_{(\mathbf{x}, \mathbf{y}) \sim \pi} c(\mathbf{x}, \mathbf{y}), \text{ s.t. } \Gamma = \{\pi : T_{\#}^{\mathbf{x}} \pi = p, T_{\#}^{\mathbf{y}} \pi = q\}, \quad (2)$$

where $c$ is the cost function. $T_{\#}^{\mathbf{x}} \pi$ is the marginal distribution of $\pi$ *w.r.t.* random variable $\mathbf{x}$. $T_{\#}^{\mathbf{x}} \pi = p$ means $\int \pi(\mathbf{x}, \mathbf{y}) d\mathbf{y} = p(\mathbf{x}), \forall \mathbf{x} \in \mathcal{X}$. Similarly, $T_{\#}^{\mathbf{y}} \pi = q$ indicates $\int \pi(\mathbf{x}, \mathbf{y}) d\mathbf{x} = q(\mathbf{y}), \forall \mathbf{y} \in \mathcal{Y}$.

**Semi-supervised OT.** The semi-supervised OT is pioneered by [28, 30]. In semi-supervised OT, a few matched pairs of source and target data points (called "keypoints") $\mathcal{K} = \{(\mathbf{x}_k, \mathbf{y}_k)\}_{k=1}^K$ are given, where $K$ is the number of keypoint pairs. The semi-supervised OT aims to leverage the given matched keypoints to guide the correct transport in OT by preserving the relation of each data point to the keypoints. Mathematically, we have

$$\min_{\tilde{\pi} \in \tilde{\Gamma}} \mathbb{E}_{(\mathbf{x}, \mathbf{y}) \sim m \otimes \tilde{\pi}} g(\mathbf{x}, \mathbf{y}), \text{ s.t. } \tilde{\Gamma} = \{\tilde{\pi} : T_{\#}^{\mathbf{x}}(m \otimes \tilde{\pi}) = p, T_{\#}^{\mathbf{y}}(m \otimes \tilde{\pi}) = q\}, \quad (3)$$

where the transport plan $m \otimes \tilde{\pi}$ is $(m \otimes \tilde{\pi})(\mathbf{x}, \mathbf{y}) = m(\mathbf{x}, \mathbf{y})\tilde{\pi}(\mathbf{x}, \mathbf{y})$, and $m$ is a binary mask function. Given a pair of keypoints $(\mathbf{x}_{k_0}, \mathbf{y}_{k_0}) \in \mathcal{K}$, then $m(\mathbf{x}_{k_0}, \mathbf{y}_{k_0}) = 1$, $m(\mathbf{x}_{k_0}, \mathbf{y}) = 0$ for $\mathbf{y} \neq \mathbf{y}_{k_0}$, and $m(\mathbf{x}, \mathbf{y}_{k_0}) = 0$ for $\mathbf{x} \neq \mathbf{x}_{k_0}$. $m(\mathbf{x}, \mathbf{y}) = 1$ if $\mathbf{x}, \mathbf{y}$ do not coincide with any keypoint. The mask-based modeling of the transport plan ensures that the keypoint pairs are always matched in the derived transport plan. $g$ in Eq. (3) is defined as $g(\mathbf{x}, \mathbf{y}) = d(R_{\mathbf{x}}^s, R_{\mathbf{y}}^t)$, where $R_{\mathbf{x}}^s, R_{\mathbf{y}}^s \in (0, 1)^K$ model the vector of relation of $\mathbf{x}, \mathbf{y}$ to each of the paired keypoints in source and target domain respectively, and $d$ is the Jensen–Shannon divergence. The $k$-th elements of $R_{\mathbf{x}}^s$ and $R_{\mathbf{x}}^s$ are respectively defined by

$$R_{\mathbf{x},k}^s = \frac{\exp(-c(\mathbf{x}, \mathbf{x}_k)/\tau)}{\sum_{l=1}^K \exp(-c(\mathbf{x}, \mathbf{x}_l)/\tau)}, \; R_{\mathbf{y},k}^t = \frac{\exp(-c(\mathbf{y}, \mathbf{y}_k)/\tau)}{\sum_{l=1}^K \exp(-c(\mathbf{y}, \mathbf{y}_l)/\tau)}, \quad (4)$$

where $\tau$ is set to 0.1. Note that, to ensure feasible solutions, the mass of paired keypoints should be equal, *i.e.*, $p(\mathbf{x}_k) = q(\mathbf{y}_k), \forall (\mathbf{x}_k, \mathbf{y}_k) \in \mathcal{K}$. Please refer to [28] for more details.

$L_2$-**regularized unsupervised and semi-supervised OTs.** As in Eqs. (2) and (3), both the unsupervised and semi-supervised OTs are linear programs that are computationally expensive to solve for larger sizes of training datasets. Researchers then present the $L_2$-regularized versions of unsupervised and semi-supervised OTs that can be solved by training networks [31, 32]. The $L_2$-regularized unsupervised and semi-supervised OTs are respectively given by

$$\min_{\pi \in \Gamma} \mathbb{E}_{(\mathbf{x}, \mathbf{y}) \sim \pi} c(\mathbf{x}, \mathbf{y}) + \epsilon \chi^2(\pi \| p \times q) \text{ and } \min_{\tilde{\pi} \in \tilde{\Gamma}} \mathbb{E}_{(\mathbf{x}, \mathbf{y}) \sim m \otimes \tilde{\pi}} g(\mathbf{x}, \mathbf{y}) + \epsilon \chi^2(m \otimes \tilde{\pi} \| p \times q), \quad (5)$$

where $\chi^2(\pi \| p \times q) = \int \frac{\pi(\mathbf{x}, \mathbf{y})^2}{p(\mathbf{x})q(\mathbf{y})} d\mathbf{x} d\mathbf{y}$, $\epsilon$ is regularization factor. The duality of the $L_2$-regularized unsupervised and semi-supervised OTs can be unified in the following formulation [31, 32]:

$$\max_{u,v} \mathcal{F}_{\text{OT}}(u, v) = \mathbb{E}_{\mathbf{x} \sim p} u(\mathbf{x}) + \mathbb{E}_{\mathbf{y} \sim q} v(\mathbf{y}) - \frac{1}{4\epsilon} \mathbb{E}_{\mathbf{x} \sim p, \mathbf{y} \sim q} I(\mathbf{x}, \mathbf{y}) \left[ (u(\mathbf{x}) + v(\mathbf{y}) - \xi(\mathbf{x}, \mathbf{y}))_+ \right]^2, \quad (6)$$

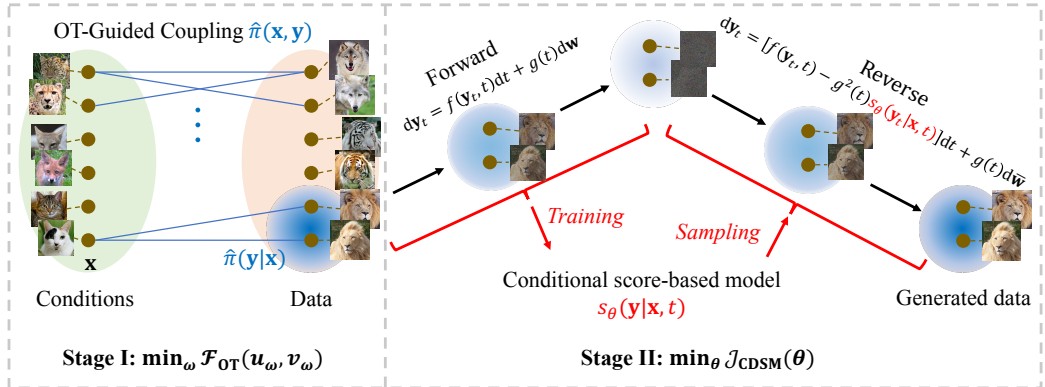

Figure 1: Illustation of optimal transport-guided conditional score-based diffusion model. We build the coupling $\hat{\pi}(\mathbf{x}, \mathbf{y})$ of condition data (*e.g.*, source images in I2I) $\mathbf{x}$ and target data $\mathbf{y}$ guided by OT. Based on the coupling, we train the conditional score-based model $s_\theta(\mathbf{y}; \mathbf{x}, t)$ by OT-guided conditional denoising score-matching that uses the forward SDE to diffuse target data to noise. With $s_\theta(\mathbf{y}; \mathbf{x}, t)$, we generate data given the condition $\mathbf{x}$ using the reverse SDE in inference.

where $a_+ = \max(a, 0)$. For unsupervised OT, $I(\mathbf{x}, \mathbf{y}) = 1$ and $\xi(\mathbf{x}, \mathbf{y}) = c(\mathbf{x}, \mathbf{y})$. For semi-supervised OT, $I(\mathbf{x}, \mathbf{y}) = m(\mathbf{x}, \mathbf{y})$ and $\xi(\mathbf{x}, \mathbf{y}) = g(\mathbf{x}, \mathbf{y})$. In [31, 32], $u, v$ are represented by neural networks $u_\omega, v_\omega$ with parameters $\omega$ that are trained by mini-batch-based stochastic optimization algorithms using the loss function in Eq. (6). The pseudo-code for training $u_\omega, v_\omega$ is given in Appendix A. Using the parameters $\hat{\omega}$ after training, the estimate of optimal transport plan is

$$\hat{\pi}(\mathbf{x}, \mathbf{y}) = H(\mathbf{x}, \mathbf{y})p(\mathbf{x})q(\mathbf{y}), \text{ where } H(\mathbf{x}, \mathbf{y}) = \frac{1}{2\epsilon}I(\mathbf{x}, \mathbf{y})\left(u_{\hat{\omega}}(\mathbf{x}) + v_{\hat{\omega}}(\mathbf{y}) - \xi(\mathbf{x}, \mathbf{y})\right)_+. \quad (7)$$

$H$ is called compatibility function. Note that $\hat{\pi}$ and $H$ depend on $c$ and $\epsilon$, which will be specified in experimental details in Appendix C.

## 3   OT-Guided Conditional SBDM

We aim to develop conditional SBDM for applications with unpaired or partially paired setting. The unpaired setting means that there is no paired condition data and target data in training. For example, in unpaired image-to-image translation (I2I), the source and target data in the training set are all unpaired. For the partially paired setting, along with the unpaired set of condition data (e.g., source data in I2I) and target data, we are also given a few paired condition data and target data.

We propose an Optimal Transport-guided Conditional SBDM, dubbed OTCS, for conditional diffusion in both unpaired and partially paired settings. The basic idea is illustrated in Fig. 1. Since the condition data and target data are not required to be paired in the training dataset, we build the coupling relationship of condition and target data using OT [27, 28]. Based on the estimated coupling, we propose the OT-guided conditional denoising score matching to train the conditional score-based model in the unpaired or partially paired setting. With the trained conditional score-based model, we generate a sample by the reverse SDE given the condition. We next elaborate on the motivations, OT-guided conditional denoising score matching, training, and inference of our approach.

### 3.1   Motivations for OT-Guided Conditional SBDM

We provide a reformulation for the conditional score-based model with paired training dataset discussed in Sect. 2.1, and this formulation motivates us to extend the conditional SBDM to unpaired and partially paired settings in Sect. 3.2. Let $q$ and $p$ respectively denote the distributions of target data and condition data. In the paired setting, we denote the condition data as $\mathbf{x}_{\text{cond}}(\mathbf{y})$ for a target data $\mathbf{y}$, and $p$ is the measure by push-forwarding $q$ using $\mathbf{x}_{\text{cond}}$, *i.e.*, $p(\mathbf{x}) = \sum_{\{\mathbf{y}:\mathbf{x}_{\text{cond}}(\mathbf{y})=\mathbf{x}\}} q(\mathbf{y})$ over the paired training dataset.

**Proposition 1.** *Let* $\mathcal{C}(\mathbf{x}, \mathbf{y}) = \frac{1}{p(\mathbf{x})}\delta(\mathbf{x}-\mathbf{x}_{\text{cond}}(\mathbf{y}))$ *where $\delta$ is the Dirac delta function, then $\mathcal{J}_{\text{DSM}}(\theta)$ in Eq. (1) can be reformulated as*

$$\mathcal{J}_{\text{DSM}}(\theta) = \mathbb{E}_t w_t \mathbb{E}_{\mathbf{x}\sim p}\mathbb{E}_{\mathbf{y}\sim q}\mathcal{C}(\mathbf{x}, \mathbf{y})\mathbb{E}_{\mathbf{y}_t\sim p_{t|0}(\mathbf{y}_t|\mathbf{y})} \left\| s_\theta(\mathbf{y}_t; \mathbf{x}, t) - \nabla_{\mathbf{y}_t} \log p_{t|0}(\mathbf{y}_t|\mathbf{y}) \right\|_2^2. \quad (8)$$

*Furthermore, $\gamma(\mathbf{x}, \mathbf{y}) = \mathcal{C}(\mathbf{x}, \mathbf{y})p(\mathbf{x})q(\mathbf{y})$ is a joint distribution for marginal distributions $p$ and $q$.*

The proof is given in Appendix B. From Proposition 1, we have the following observations. First, the coupling relationship of condition data and target data is explicitly modeled in $\mathcal{C}(\mathbf{x}, \mathbf{y})$. Second, the joint distribution $\gamma$ exhibits a similar formulation to the transport plan $\hat{\pi}$ in Eq. (7). The definition of $\mathcal{C}(\mathbf{x}, \mathbf{y})$ in Proposition 1 is for paired $\mathbf{x}, \mathbf{y}$. While for the unpaired or partially paired setting, the definition of $\mathcal{C}(\mathbf{x}, \mathbf{y})$ is not obvious due to the lack of paired relationship between $\mathbf{x}, \mathbf{y}$. We therefore consider modeling the joint distribution of condition data ($\mathbf{x}$) and target data ($\mathbf{y}$) by $L_2$-regularized OT (see Sect. 2.2) for the unpaired and partially paired settings, in which the coupling relationship of condition data and target data is built in the compatibility function $H(\mathbf{x}, \mathbf{y})$.

### 3.2  OT-Guided Conditional Denoising Score Matching

With the motivations discussed in Sects. 1 and 3.1, we model the coupling relationship between condition data and target data for unpaired and partially paired settings using $L_2$-regularized unsupervised and semi-supervised OTs, respectively. Specifically, the $L_2$-regularized unsupervised and semi-supervised OTs are applied to the distributions $p, q$, and the coupling relationship of the condition data $\mathbf{x}$ and target data $\mathbf{y}$ is built by the compatibility function $H(\mathbf{x}, \mathbf{y})$. We then extend the formulation for paired setting in Eq. (8) by replacing $\mathcal{C}$ with $H$ to develop the training objective for unpaired and partially paired settings, which is given by

$$\mathcal{J}_{\text{CDSM}}(\theta) = \mathbb{E}_t w_t \mathbb{E}_{\mathbf{x}\sim p}\mathbb{E}_{\mathbf{y}\sim q}H(\mathbf{x}, \mathbf{y})\mathbb{E}_{\mathbf{y}_t\sim p_{t|0}(\mathbf{y}_t|\mathbf{y})} \left\| s_\theta(\mathbf{y}_t; \mathbf{x}, t) - \nabla_{\mathbf{y}_t} \log p_{t|0}(\mathbf{y}_t|\mathbf{y}) \right\|_2^2. \quad (9)$$

Equation (9) is dubbed "OT-guided conditional denoising score matching". In Eq. (9), $H$ is a "soft" coupling relationship of condition data and target data, because there may exist multiple $\mathbf{x}$ satisfying $H(\mathbf{x}, \mathbf{y}) > 0$ for each $\mathbf{y}$. While Eq. (8) assumes "hard" coupling relationship, *i.e.*, there is only one condition data $\mathbf{x}$ for each $\mathbf{y}$ satisfying $\mathcal{C}(\mathbf{x}, \mathbf{y}) > 0$. We minimize $\mathcal{J}_{\text{CDSM}}(\theta)$ to train the conditional score-based model $s_\theta(\mathbf{y}_t; \mathbf{x}, t)$. We will theoretically analyze that our formulation in Eq. (9) is still a diffusion model in Sect. 3.5, and empirically compare the "soft" and "hard" coupling relationship in Appendix D.

### 3.3  Training the Conditional Score-based Model

To implement $\mathcal{J}_{\text{CDSM}}(\theta)$ in Eq. (9) using training samples to optimize $\theta$, we can sample mini-batch data $\boldsymbol{X}$ and $\boldsymbol{Y}$ from $p$ and $q$ respectively, and then compute $H(\mathbf{x}, \mathbf{y})$ and $\mathcal{J}_{\mathbf{x},\mathbf{y}} = \mathbb{E}_t w_t \mathbb{E}_{\mathbf{y}_t\sim p_{t|0}(\mathbf{y}_t|\mathbf{y})}$ $\|s_\theta(\mathbf{y}_t; \mathbf{x}, t) - \nabla_{\mathbf{y}_t} \log p_{t|0}(\mathbf{y}_t|\mathbf{y})\|_2^2$ over the pairs of $(\mathbf{x}, \mathbf{y})$ in $\boldsymbol{X}$ and $\boldsymbol{Y}$. However, such a strategy is sub-optimal. This is because given a mini-batch of samples $\boldsymbol{X}$ and $\boldsymbol{Y}$, for each source sample $\mathbf{x}$, there may not exist target sample $\mathbf{y}$ in the mini-batch with a higher value of $H(\mathbf{x}, \mathbf{y})$ that matches condition data $\mathbf{x}$. Therefore, few or even no samples in a mini-batch contribute to the loss function in Eq. (9), leading to a large bias of the computed loss and instability of the training. To tackle this challenge, we propose a "resampling-by-compatibility" strategy to compute the loss in Eq. (9).

**Resampling-by-compatibility.**   To implement the loss in Eq. (9), we perform the following steps:

- Sample $\mathbf{x}$ from $p$ and sample $\boldsymbol{Y}_{\mathbf{x}} = \{\mathbf{y}^l\}_{l=1}^L$ from $q$;
- Resample a $\mathbf{y}$ from $\boldsymbol{Y}_{\mathbf{x}}$ with the probability proportional to $H(\mathbf{x}, \mathbf{y}^l)$;
- Compute the training loss $\mathcal{J}_{\mathbf{x},\mathbf{y}}$ in the above paragraph on the sampled pair $(\mathbf{x}, \mathbf{y})$.

In implementation, $u_\omega$ and $v_\omega$ (introduced above Eq. (7) in Sect. 2.2) are often lightweight neural networks, so $H(\mathbf{x}, \mathbf{y}^l)$ can be computed fast as in Eq. (7). In the applications where we are given training datasets of samples, for each $\mathbf{x}$ in the set of condition data, we choose all the samples $\mathbf{y}$ in the set of target data satisfying $H(\mathbf{x}, \mathbf{y}) > 0$ to construct $\boldsymbol{Y}_{\mathbf{x}}$, and meanwhile store the corresponding values of $H(\mathbf{x}, \mathbf{y})$. This is done before training $s_\theta$. During training, we directly choose $\mathbf{y}$ from $\boldsymbol{Y}_{\mathbf{x}}$ based on the stored values of $H$, which speeds up the training process. Please refer to Appendix A for the rationality of the resampling-by-compatibility.

**Algorithm.**  Before training $s_\theta$, we train $u_\omega, v_\omega$ using the duality of $L_2$-regularized unsupervised and semi-supervised OTs in Eq. (6) for unpaired and partially paired settings, respectively. During training $s_\theta$, in each iteration, we sample a mini-batch of condition data $\boldsymbol{X} = \{\mathbf{x}_b\}_{b=1}^B$ from $p$ with batch size $B$. For $\mathbf{x}_b$, we sequentially obtain a sample $\mathbf{y}$ using our resampling-by-compatibility strategy, uniformly sample a $t$ in $[0, T]$, and generate a noisy data $\mathbf{y}_t$ from $p_{t|0}(\mathbf{y}_t|\mathbf{y})$. We then compute the value of $\mathcal{J}_{\mathbf{x}_b, \mathbf{y}}$ (defined in the first paragraph of this section) on $t$ and $\mathbf{y}_t$, which is averaged over all $b$ as the final training loss. The parameters $\theta$ of $s_\theta$ are then updated by stochastic optimization algorithms, *e.g.*, Adam. The pseudo-code of the training algorithm is given in Appendix A.

### 3.4  Sample Generation

We denote the trained conditional score-based model as $s_{\hat{\theta}}(\mathbf{y}; \mathbf{x}, t)$ where $\hat{\theta}$ is the value of $\theta$ after training. Given the condition data $\mathbf{x}$, we generate target samples by the following SDE:

$$\mathrm{d}\mathbf{y}_t = \left[ f(\mathbf{y}_t, t) - g(t)^2 s_{\hat{\theta}}(\mathbf{y}_t; \mathbf{x}, t) \right] \mathrm{d}t + g(t)\,\mathrm{d}\bar{\mathbf{w}}, \tag{10}$$

with the initial state $\mathbf{y}_T \sim p_{\mathrm{prior}}$. The numerical SDE solvers, *e.g.*, Euler-Maruyama method [33], DDIM [34], and DPM-Solver [35], are then applied to solve the above reverse SDE.

### 3.5  Analysis

In this section, we analyze that by Eq. (10), our approach approximately generates samples from the conditional transport plan $\hat{\pi}(\mathbf{y}|\mathbf{x})$, where $\hat{\pi}(\mathbf{y}|\mathbf{x}) = H(\mathbf{x}, \mathbf{y})q(\mathbf{y})$ is based on $\hat{\pi}(\mathbf{x}, \mathbf{y})$ in Eq. (7).

**Theorem 1.** *For $\mathbf{x} \sim p$, we define the forward SDE $\mathrm{d}\mathbf{y}_t = f(\mathbf{y}_t, t)\,\mathrm{d}t + g(t)\,\mathrm{d}\mathbf{w}$ with $\mathbf{y}_0 \sim \hat{\pi}(\cdot|\mathbf{x})$ and $t \in [0, T]$, where $f, g, T$ are given in Sect. 2.1. Let $p_t(\mathbf{y}_t|\mathbf{x})$ be the corresponding distribution of $\mathbf{y}_t$ and $\mathcal{J}_{\mathrm{CSM}}(\theta) = \mathbb{E}_t w_t \mathbb{E}_{\mathbf{x} \sim p} \mathbb{E}_{\mathbf{y}_t \sim p_t(\mathbf{y}_t|\mathbf{x})} \|s_\theta(\mathbf{y}_t; \mathbf{x}, t) - \nabla_{\mathbf{y}_t} \log p_t(\mathbf{y}_t|\mathbf{x})\|_2^2$, then we have $\nabla_\theta \mathcal{J}_{\mathrm{CDSM}}(\theta) = \nabla_\theta \mathcal{J}_{\mathrm{CSM}}(\theta)$.*

We give the proof in Appendix B. Theorem 1 indicates that the trained $s_\theta(\mathbf{y}_t; \mathbf{x}, t)$ using Eq. (9) approximates $\nabla_{\mathbf{y}_t} \log p_t(\mathbf{y}_t|\mathbf{x})$. Based on Theorem 1, we can interpret our approach as follows. Given a condition data $\mathbf{x}$, we sample target data $\mathbf{y}_0$ from the conditional transport plan $\hat{\pi}(\mathbf{y}_0|\mathbf{x})$, produce $\mathbf{y}_t$ by the forward SDE, and train $s_\theta(\mathbf{y}_t; \mathbf{x}, t)$ to approximate $\nabla_{\mathbf{y}_t} \log p_t(\mathbf{y}_t|\mathbf{x})$, as illustrated in Fig. 1. This implies that Eq. (10) approximates the reverse SDE $\mathrm{d}\mathbf{y}_t = [f(\mathbf{y}_t, t) - g(t)^2 \nabla_{\mathbf{y}_t} \log p_t(\mathbf{y}_t|\mathbf{x})]\,\mathrm{d}t + g(t)\,\mathrm{d}\bar{\mathbf{w}}$, by which the generated samples at time $t = 0$ are from $p_0(\mathbf{y}|\mathbf{x}) = \hat{\pi}(\mathbf{y}|\mathbf{x})$ given the initial state $\mathbf{y}_T \sim p_T(\mathbf{y}_T|\mathbf{x})$.

## 4  OTCS Realizes Data Transport for Optimal Transport

As discussed in Sect. 3, OTCS is proposed to learn the conditional SBDM guided by OT. In this section, we will show that, from the viewpoint of OT, OTCS offers a diffusion-based approach to transport data for OT. Given a source distribution $p(\mathbf{x})$ and a target distribution $q(\mathbf{y})$, the derived coupling $\pi(\mathbf{x}, \mathbf{y})$ models the joint probability density function rather than the transported sample of $\mathbf{x}$. How to transport the source data points to the target domain is known to be a challenging problem for large-scale OT [31, 36]. Based on $\pi$, Seguy *et al.* [31] transport $\mathbf{x}$ to the barycenter of $\pi(\cdot|\mathbf{x})$ which is a blurred sample. Daniels *et al.* [26] transport $\mathbf{x}$ to a target sample generated from $\pi(\cdot|\mathbf{x})$. In line with [26], we next theoretically show that our proposed OTCS can generate samples from $\pi(\cdot|\mathbf{x})$. The comparison of OTCS with [26] will be given in the last paragraph of this section.

We next study the upper bound of the distance between the distribution (denoted as $p^{\mathrm{sde}}(\mathbf{y}|\mathbf{x})$) of generated samples by OTCS and the optimal conditional transport plan $\pi(\mathbf{y}|\mathbf{x})$. For convenience, we investigate the upper bound of the expected Wasserstein distance $\mathbb{E}_{\mathbf{x} \sim p} W_2\left(p^{\mathrm{sde}}(\cdot|\mathbf{x}), \pi(\cdot|\mathbf{x})\right)$. We denote the Lagrange function for the $L_2$-regularized unsupervised or semi-supervised OTs in Eq. (6) as $\mathcal{L}(\pi, u, v)$ with dual variables $u, v$ as follows:

$$\begin{aligned}
\mathcal{L}(\pi, u, v) &= \int \left( \xi(\mathbf{x}, \mathbf{y})\pi(\mathbf{x}, \mathbf{y}) + \epsilon \frac{\pi(\mathbf{x}, \mathbf{y})^2}{p(\mathbf{x})q(\mathbf{y})} \right) \mathrm{d}\mathbf{x}\,\mathrm{d}\mathbf{y} \\
&+ \int u(\mathbf{x}) \left( \int \pi(\mathbf{x}, \mathbf{y})\,\mathrm{d}\mathbf{y} - p(\mathbf{x}) \right) \mathrm{d}\mathbf{x} + \int v(\mathbf{y}) \left( \int \pi(\mathbf{x}, \mathbf{y})\,\mathrm{d}\mathbf{x} - q(\mathbf{y}) \right) \mathrm{d}\mathbf{y}.
\end{aligned} \tag{11}$$

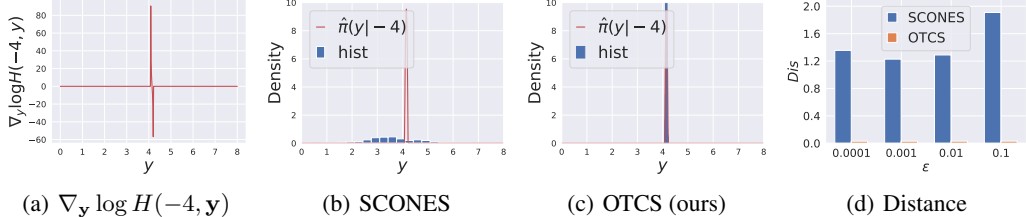

| (a) $\nabla_{\mathbf{y}} \log H(-4, \mathbf{y})$ | (b) SCONES | (c) OTCS (ours) | (d) Distance |

Figure 2: 1-D example of $L_2$-regularized unsupervised OT between $p(\mathbf{x}) = \mathcal{N}(\mathbf{x}|-4, 1)$ and $q(\mathbf{y}) = \mathcal{N}(\mathbf{y}|4, 1)$. (a) The gradient $\nabla_{\mathbf{y}} \log H(-4, \mathbf{y})$. (b-c) The conditional transport plan $\hat{\pi}(\mathbf{y}|-4)$ and the histograms of generated samples by (b) SCONES and (c) OTCS with $\epsilon = 0.0001$. (d) The distance $Dis = \mathbb{E}_{\mathbf{x} \sim p} W_2 \left( p^{\text{sde}}(\cdot|\mathbf{x}), \pi(\cdot|\mathbf{x}) \right)$ for different $\epsilon$ computed using samples in which we approximate $p^{\text{sde}}(\cdot|\mathbf{x})$ and $\pi(\cdot|\mathbf{x})$ by Gaussian distributions with corresponding mean and variance.

For semi-supervised OT, $\pi$ is further constrained by $\pi = m \otimes \tilde{\pi}$. We follow [26] to assume that $\mathcal{L}(\pi, u, v)$ is $\kappa$-strongly convex in $L_1$-norm *w.r.t.* $\pi$, and take the assumptions (see Appendix B) in [37] that investigates the bound for unconditional SBDMs.

**Theorem 2.** *Suppose the above assumption and the assumptions in Appendix B hold, and $w_t = g(t)^2$, then we have*

$$
\begin{aligned}
\mathbb{E}_{\mathbf{x} \sim p} W_2 \left( p^{\text{sde}}(\cdot|\mathbf{x}), \pi(\cdot|\mathbf{x}) \right) \leq & C_1 \left\| \nabla_{\hat{\pi}} \mathcal{L}(\hat{\pi}, u_{\hat{\omega}}, v_{\hat{\omega}}) \right\|_1 + \sqrt{C_2 \mathcal{J}_{\text{CSM}}(\hat{\theta})} \\
& + C_3 \mathbb{E}_{\mathbf{x} \sim p} W_2 \left( p_T(\cdot|\mathbf{x}), p_{\text{prior}} \right),
\end{aligned}
\tag{12}
$$

*where $C_1, C_2$, and $C_3$ are constants to $\hat{\omega}$ and $\hat{\theta}$ given in Appendix B.*

The proof is provided in Appendix B. In OTCS, we use $u_{\omega}$ and $v_{\omega}$ to respectively parameterize $u$ and $v$ as discussed in Sect. 2.2. The trained $u_{\hat{\omega}}$ and $v_{\hat{\omega}}$ are the minimizer of the dual problem in Eq. (6) that are near to the saddle point of $\mathcal{L}(\pi, u, v)$. This implies that the gradient norm $\left\| \nabla_{\hat{\pi}} \mathcal{L}(\hat{\pi}, u_{\hat{\omega}}, v_{\hat{\omega}}) \right\|_1$ of the Lagrange function *w.r.t.* the corresponding primal variable $\hat{\pi}$ is minimized in our approach. The conditional score-based model $s_{\theta}$ is trained to minimize $\mathcal{J}_{\text{CSM}}(\theta)$ according to Theorem 1. So the loss $\mathcal{J}_{\text{CSM}}(\hat{\theta})$ of trained $s_{\hat{\theta}}$ is minimized. We choose the forward SDE such that $p_T(\cdot|\mathbf{x})$ is close to $p_{\text{prior}}$, which minimizes $\mathbb{E}_{\mathbf{x} \sim p} W_2 \left( p_T(\cdot|\mathbf{x}), p_{\text{prior}} \right)$. Therefore, OTCS minimizes $\mathbb{E}_{\mathbf{x} \sim p} W_2 \left( p^{\text{sde}}(\cdot|\mathbf{x}), \pi(\cdot|\mathbf{x}) \right)$, indicating that OTCS can approximately generate samples from $\pi(\cdot|\mathbf{x})$.

**Comparison with related large-scale OT methods.** Recent unsupervised OT methods [26, 38–40] often parameterize the transport map by a neural network learned by adversarial training based on the dual formulation [38–40], or generate samples from the conditional transport plan [26]. Our method is mostly related to SCONES [26] that leverages the unconditional SBDM for generating samples from the estimated conditional transport plan $\hat{\pi}(\mathbf{y}|\mathbf{x})$. Motivated by the expression of $\hat{\pi}(\mathbf{y}|\mathbf{x})$ in Sect. 3.5, given source sample $\mathbf{x}$, SCONES generates target sample $\mathbf{y}$ by the reverse SDE $d\mathbf{y}_t = [f(\mathbf{y}_t, t) - g(t)^2 \left( \nabla_{\mathbf{y}_t} \log H(\mathbf{x}, \mathbf{y}_t) + s_{\theta}(\mathbf{y}_t; t) \right)] dt + g(t) d\bar{\mathbf{w}}$, where $s_{\theta}(\mathbf{y}_t; t)$ is the unconditional score-based model. The compatibility function $H$ is trained on clean data. While in SCONES, $\nabla_{\mathbf{y}_t} \log H(\mathbf{x}, \mathbf{y}_t)$ is computed on noisy data $\mathbf{y}_t$ for $t > 0$. OTCS computes $H$ on clean data as in Eq. (9). We provide a 1-D example in Fig. 2 to evaluate SCONES and OTCS. From Figs. 2(b-c), we can see that the sample histogram, generated by OTCS given "-4" as condition, better fits $\hat{\pi}(\cdot|-4)$. In Fig. 2(d), OTCS achieves a lower expected Wasserstein distance $\mathbb{E}_{\mathbf{x} \sim p} W_2 \left( p^{\text{sde}}(\cdot|\mathbf{x}), \hat{\pi}(\cdot|\mathbf{x}) \right)$. In SCONES, the gradient of $H$ on noisy data could be inaccurate or zero (as shown in Fig. 2(a)), which may fail to guide the noisy data $\mathbf{y}_t$ to move towards the desired locations as $t \to 0$. Our OTCS utilizes $H$ to guide the training of the conditional score-based model, without requiring the gradient of $H$ in inference. This may account for the better performance of OTCS.

## 5 Experiments

We evaluate OTCS on unpaired super-resolution and semi-paired I2I. Due to space limits, additional experimental details and results are given in Appendix C and D, respectively.

Table 1: Quantitative results for unpaired super-resolution on Celeba and semi-paired I2I on Animal images and Digits. The best and second best are respectively bolded and underlined.

| Method | Method Type | Celeba | | Animal images | | Digits | |
|---|---|---|---|---|---|---|---|
| | | FID ↓ | SSIM ↑ | FID ↓ | Acc (%) ↑ | FID ↓ | Acc (%) ↑ |
| W2GAN [42] | OT | 48.83 | 0.7169 | 118.45 | 33.56 | 97.06 | 29.13 |
| OT-ICNN [38] | OT | 33.26 | 0.8904 | 148.29 | 38.44 | 50.33 | 10.84 |
| OTM [39] | OT | 22.93 | 0.8302 | 69.27 | 33.11 | 18.67 | 9.48 |
| NOT [43] | OT | 13.65 | 0.9157 | 156.07 | 28.44 | 23.90 | 15.72 |
| KNOT [40] | OT | 5.95 | 0.8887 | 118.26 | 27.33 | **3.18** | 9.25 |
| ReFlow [45] | Flow | 70.69 | 0.4544 | 56.04 | 29.33 | 138.59 | 11.57 |
| EGSDE [20] | Diffusion | 11.49 | 0.3835 | 52.11 | 29.33 | 34.72 | 11.78 |
| DDIB [44] | Diffusion | 11.35 | 0.1275 | 28.45 | 32.44 | 9.47 | 9.15 |
| TCR [17] | – | – | – | 34.61 | 40.44 | 6.90 | 36.21 |
| SCONES [26] | OT, Diffusion | 15.46 | 0.1042 | 25.24 | 35.33 | 6.68 | 10.37 |
| **OTCS (ours)** | OT, Diffusion | **1.77** | **0.9313** | **13.68** | **96.44** | 5.12 | **67.42** |

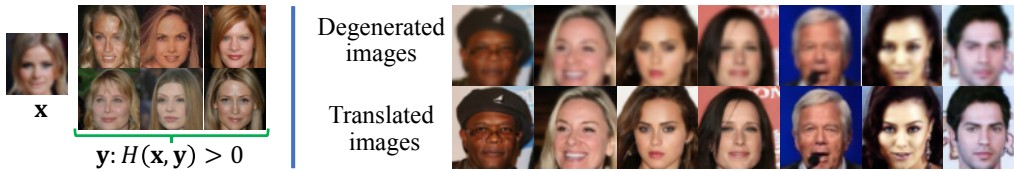

Figure 3: Left: The guided high-resolution images (*i.e.*, $\mathbf{y} : H(\mathbf{x}, \mathbf{y}) > 0$) sampled based on OT for low-resolution image $\mathbf{x}$ in training. Right: results of OTCS on CelebA (64×64) in unpaired setting.

## 5.1 Unpaired Super-Resolution on CelebA Faces

We consider the unpaired super-resolution for 64×64 aligned faces on CelebA dataset [41]. Following [26], we split the dataset into 3 disjointed subsets: A1 (90K), B1 (90K), and C1 (30K). For images in each subset, we do 2× bilinear downsampling to obtain the low-resolution images, followed by 2× bilinear upsampling to generate datasets of A0, B0, and C0 correspondingly. We train the model on A0 and B1 respectively as the source and target datasets, and test on C0 for generating high-resolution images. To apply OTCS to this task, we take B1 as target data and A0 as unpaired source data. We use the $L_2$-regularized unsupervised OT to estimate the coupling, where $c$ is set to the mean squared $L_2$-distance. In testing, given a degenerated image $\mathbf{x}$ in C0 as condition, we follow [19, 20] to sample noisy image $\mathbf{y}_M$ from $p_{M|0}(\mathbf{y}|\mathbf{x})$ as initial state and perform the reverse SDE to generate the high-resolution image. $M = 0.2$ in experiments. Our approach is compared with recent adversarial-training-based unsupervised OT methods [38–40, 42, 43], diffusion-based unpaired I2I methods [20, 44], flow-based I2I method [45], and SCONES [26] (discussed in Sect. 4). We use the FID score [46] to measure the quality of translated images, and the SSIM metric to measure the structural similarity of each translated image to its ground-truth high-resolution image in C1.

In Tab. 1, OTCS achieves the lowest FID and the highest SSIM on CelebA dataset among the compared methods. We can observe that in general, the adversarial-training-based OT methods [38–40, 42, 43] achieve better SSIM than the diffusion-based methods [20, 26, 44, 45]. This could be because the OT guidance is imposed in training by these OT methods. By contrast, the diffusion-based methods [20, 26, 44, 45] generally achieve better FID than OT methods [38–40, 42, 43], which may be attributed to the capability of diffusion models for generating high-quality images. Our OTCS imposes the OT guidance in the training of diffusion models, integrating both advantages of OT and diffusion models. In Fig. 3, we show the guided high-resolution images sampled based on OT (left) and translated images by OTCS (right). We also report the FID (0.78) and SSIM (0.9635) of the Oracle that uses true high-resolution images in A1 as paired data for training. We can see that OTCS approaches the Oracle.

## 5.2 Semi-paired Image-to-Image Translation

We consider the semi-paired I2I task that a large number of unpaired along with a few paired images across source and target domains are given for training. The goal of semi-paired I2I is to leverage the

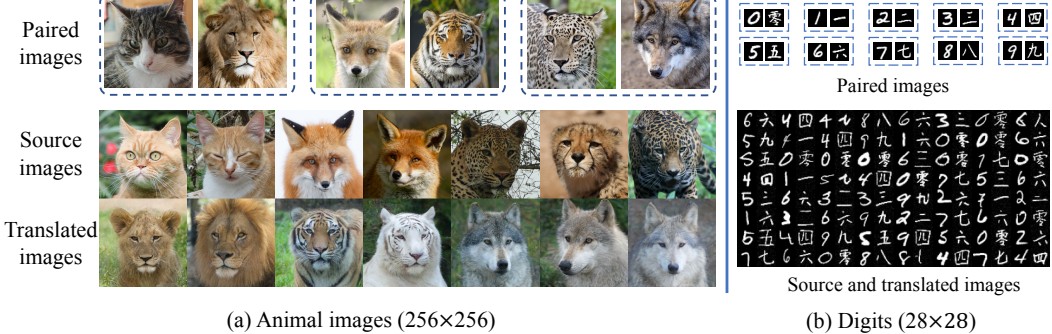

Paired images

Source images

Translated images

Paired images

Source and translated images

(a) Animal images (256×256)       (b) Digits (28×28)

Figure 4: Results of OTCS for semi-paired I2I on (a) Animal images and (b) Digits. The bottom of Fig. 4(b) plots source (odd columns) and corresponding translated (even columns) images.

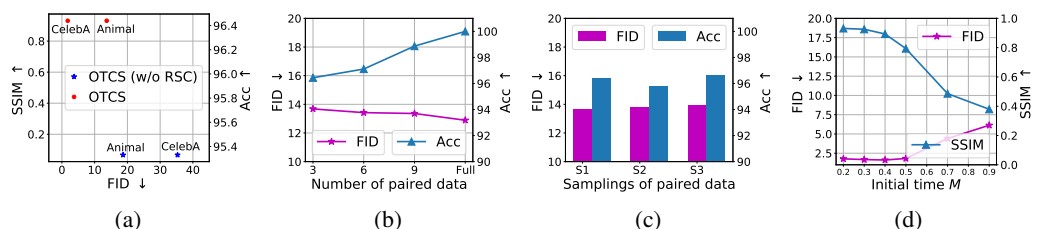

(a)      (b)      (c)      (d)

Figure 5: (a) Results of OTCS w/ and w/o RSC. (b-c) Results of OTCS with varying (b) numbers and (c) samplings of paired data on Animal images. (d) Results of OTCS for different $M$ on CelebA.

paired cross-domain images to guide the desired translation of source images to the target domain. To apply our approach to the semi-paired I2I, we take the source images as condition data. We use the $L_2$-regularized semi-supervised OT to estimate the coupling. The cost $c$ and $c'$ in Eq. (4) are taken as the cosine dissimilarity of features using the image encoder of CLIP [47]. Apart from the compared unpaired I2I approaches in Sect. 5.1, we additionally compare our approach with the semi-paired I2I approach TCR [17]. For the OT-based approaches [38–40, 42, 43], we additionally impose the matching of the paired images using a reconstruction loss in them for semi-paired I2I. It is non-trivial to impose the matching of the paired images in the diffusion/flow-based approaches [20, 44, 45]. We adopt two metrics of FID and Accuracy (Acc). The FID measures the quality of generated samples. The higher Acc implies that the guidance of the paired images is better realized for desired translation. The experiments are conducted on digits and natural animal images.

For *Animal images*, we take images of cat, fox, and leopard from AFHQ [48] dataset as source, and images of lion, tiger, and wolf as target. Three cross-domain image pairs are given, as shown in Fig. 4(a). By the guidance of the paired images, we expect that the cat, fox, and leopard images are respectively translated to the images of lion, tiger, and wolf. The Acc is the ratio of source images translated to ground-truth translated classes (GTTCs), where the GTTCs of source images of cat/fox/leopard are lion/tiger/wolf. For *Digits*, we consider the translation from MNIST [49] to Chinese-MNIST [50]. The MNIST and Chinese-MNIST contain the digits (from 0 to 9) in different modalities. We annotate 10 cross-domain image pairs, each corresponding to a digit, as in Fig. 4(b). With the guidance of the paired images, we expect that the source images are translated to the target ones representing the same digits. The GTTCs for the source images are their corresponding digits.

We can observe in Tab. 1 that OTCS achieves the highest Acc on both Animal images and Digits, outperforming the second-best method TCR [17] by more than 50% and 30% on the two datasets, respectively. Our approach explicitly models the guidance of the paired images to the unpaired ones using the semi-supervised OT in training, and better translates the source images to target ones of the desired classes. In terms of the FID, OTCS achieves competitive (on Digits) or even superior (on Animal images) results over the other approaches, indicating that OTCS can generate images of comparable quality. The translated images in Figs. 4 also show the effectiveness of OTCS.

### 5.3 Analysis

**Effectiveness of resampling-by-compatibility (RSC).** Figure 5(a) shows that OTCS achieves better results (left top points are better) than OTCS (w/o RSC) in semi-paired I2I on Animal images and unpaired super-resolution on CelebA, demonstrating the effectiveness of resampling-by-compatibility.

**Results on semi-paired I2I with varying numbers and samplings of paired images.** We study the effect of the number and choice of paired images in semi-paired I2I. Figure 5(b) shows that as the number of paired data increases, the Acc increases, and the FID marginally decreases. This implies that our approach can impose the guidance of different amounts of paired data to translate source images to desired classes. From Fig. 5(c), OTCS achieves similar FID and Acc for three different samplings of the same number, *i.e.*, 3, of paired images (denoted as "S1", "S2", and "S3" in Fig. 5(c)).

**Results on unpaired super-resolution with varying initial time.** We show the results of OTCS under different initial time $M$ in the reverse SDE. Figure 5(d) indicates that with a smaller $M$, our OTCS achieves better SSIM. This makes sense because adding smaller-scale noise in inference could better preserve the structure of source data in SBDMs, as in [19, 20]. However, smaller $M$ may lead to a larger distribution gap/FID between generated and target data for SBDMs with unconditional score-based model [19, 20]. We observe that OTCS achieves the FID below 1.85 for $M$ in [0.2, 0.5].

## 6 Conclusion

This paper proposes a novel Optimal Transport-guided Conditional Score-based diffusion model (OTCS) for image translation with unpaired or partially paired training dataset. We build the coupling of the condition data and target data using $L_2$-regularized unsupervised and semi-supervised OTs, and present the OT-guided conditional denoising score-matching and resampling-by-compatibility to train the conditional score-based model. Extensive experiments in unpaired super-resolution and semi-paired I2I tasks demonstrated the effectiveness of OTCS for achieving desired translation of source images. We theoretically analyze that OTCS realizes data transport for OT. In the future, we are interested in more applications of OTCS, such as medical image translation/synthesis.

## Limitations

The cost function should be determined first when using our method. In experiments, we simply choose the squared $L_2$-distance in image space for unpaired super-resolution and cosine distance in feature space for semi-paired I2I, achieving satisfactory performance. However, the performance may be improved if more domain knowledge is employed to define the cost function. Meanwhile, if the number of target data is small, the generation ability of our trained model may be limited.

## Acknowledgement

This work was supported by National Key R&D Program 2021YFA1003002 and NSFC (12125104, U20B2075, 61721002).

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
