# Optimal Transport-Guided Conditional Score-Based Diffusion Model

## (Appendix)

**Xiang Gu[1], Liwei Yang[1], Jian Sun (✉)[1,2,3], Zongben Xu[1,2,3]**
[1] School of Mathematics and Statistics, Xi'an Jiaotong University, Xi'an, China
[2] Pazhou Laboratory (Huangpu), Guangzhou, China
[3] Peng Cheng Laboratory, Shenzhen, China
{xianggu,yangliwei}@stu.xjtu.edu.cn {jiansun,zbxu}@xjtu.edu.cn

## A  Additional Details for Sections 2 and 3

### A.1  Additional Details for Section 2

**Details of VP-SDE and VE-SDE.**   As mentioned in Sect. 2.1, we choose the VE-SDE and the VP-SDE as the forward SDEs. For VE-SDE, $f(\mathbf{y}, t) = 0$ and $g(t) = \alpha^t$ where $\alpha$ is a hyper-parameter. For VP-SDE, $f(\mathbf{y}, t) = -\frac{1}{2}\beta(t)\mathbf{y}$ and $g(t) = \sqrt{\beta(t)}$ where $\beta(t) = \beta_{\min} + (\beta_{\max} - \beta_{\min})t$, and $\beta_{\min}$ and $\beta_{\max}$ are hyper-parameters. Then, the conditional distribution, *a.k.a.*, permutation kernel, $p_{t|0}(\mathbf{y}_t|\mathbf{y}_0)$ of $\mathbf{y}_t$ given $\mathbf{y}_0$ is

$$p_{t|0}(\mathbf{y}_t|\mathbf{y}_0) = \begin{cases} \mathcal{N}\left(\mathbf{y}_t|\mathbf{y}_0, \frac{1}{2\log\alpha}(\alpha^{2t} - 1)\mathbf{I}\right), & \text{for VE-SDE,} \\ \mathcal{N}\left(\mathbf{y}_t|\mathbf{y}_0 e^{\frac{1}{2}h(t)}, (1 - e^{h(t)})\mathbf{I}\right), & \text{for VP-SDE,} \end{cases} \tag{A-1}$$

where $h(t) = -\frac{1}{2}t^2(\beta_{\max} - \beta_{\min}) - t\beta_{\min}$, and $\mathbf{I}$ is the identity matrix. Following [1], we set $T$ to 1, and $p_{\text{prior}} = \mathcal{N}(0, \mathbf{I})$ for VP-SDE and $p_{\text{prior}} = \mathcal{N}(0, \frac{1}{2\log\alpha}(\alpha^{2t} - 1)\mathbf{I})$ for VE-SDE.

**Pseudo-codes of algorithm for training $u_\omega, v_\omega$.**   The pseudo-codes of the algorithm to learn the dual variables $u_\omega, v_\omega$, *a.k.a.*, potentials, are given in Algorithm 1.

---

**Algorithm 1:** Algorithm for estimating potentials $u_{\hat{\omega}}, v_{\hat{\omega}}$

---

**Input:** Distribution $p$ of conditions, target data distribution $q$, paired data (if available)
**Output:** Learned potentials $u_{\hat{\omega}}, v_{\hat{\omega}}$
**for** iter $= 1, \cdots, N'_{\text{iter}}$ **do**
    Sampling mini-batch data $\mathbf{X} = \{\mathbf{x}_b\}_{b=1}^{B'}$ from $p$, $\mathbf{Y} = \{\mathbf{y}_b\}_{b=1}^{B'}$ from $q$;
    **if** *paired data are available* **then**
        Computing the loss of semi-supervised OT in Eq. (6) on $\mathbf{X}$ and $\mathbf{Y}$;
    **else**
        Computing the loss of unsupervised OT in Eq. (6) on $\mathbf{X}$ and $\mathbf{Y}$;
    **end**
    Backward propagation to compute the gradient and update $\omega$ using Adam algorithm;
**end**
$\hat{\omega} = \omega$.

---

37th Conference on Neural Information Processing Systems (NeurIPS 2023).

## A.2 Additional Details for Section 3

**Rationality of the resampling-by-compatibility.**   We next explain the rationality of the resampling-by-compatibility presented in Sect. 3.3. For the convenience of description, for any $\mathbf{x}, \mathbf{y}$, we denote

$$\mathcal{J}_{\mathbf{x},\mathbf{y}} = \mathbb{E}_t w_t \mathbb{E}_{\mathbf{y}_t \sim p_{t|0}(\mathbf{y}_t|\mathbf{y})} \|s_\theta(\mathbf{y}_t; \mathbf{x}, t) - \nabla_{\mathbf{y}_t} \log p_{t|0}(\mathbf{y}_t|\mathbf{y})\|_2^2. \tag{A-2}$$

The training loss $\mathcal{J}_{\mathrm{CDSM}}(\theta)$ in Eq. (9) can be written as

$$\mathcal{J}_{\mathrm{CDSM}}(\theta) = \mathbb{E}_{\mathbf{x} \sim p} \mathbb{E}_{\mathbf{y} \sim q} H(\mathbf{x}, \mathbf{y}) \mathcal{J}_{\mathbf{x},\mathbf{y}}. \tag{A-3}$$

By the resampling-by-compatibility, $q$ is approximated based on samples $\mathbf{Y}_\mathbf{x}$ by $q(\mathbf{y}) \approx \frac{1}{L} \sum_{l=1}^L \delta(\mathbf{y} - \mathbf{y}^l)$. We then have

$$
\begin{aligned}
\mathcal{J}_{\mathrm{CDSM}}(\theta) &\approx \mathbb{E}_{\mathbf{x} \sim p} \frac{1}{L} \sum_{l=1}^L H(\mathbf{x}, \mathbf{y}^l) \mathcal{J}_{\mathbf{x},\mathbf{y}^l} \\
&\propto \mathbb{E}_{\mathbf{x} \sim p} \frac{1}{H_0} \sum_{l=1}^L H(\mathbf{x}, \mathbf{y}^l) \mathcal{J}_{\mathbf{x},\mathbf{y}^l} \\
&= \mathbb{E}_{\mathbf{x} \sim p} \mathbb{E}_{\mathbf{y} \sim \tilde{h}_\mathbf{x}} \mathcal{J}_{\mathbf{x},\mathbf{y}},
\end{aligned}
\tag{A-4}
$$

where $H_0 = \sum_{l=1}^L H(\mathbf{x}, \mathbf{y}^l)$ and $\tilde{h}_\mathbf{x}$ is the distribution defined on $\mathbf{Y}_\mathbf{x}$ as $\tilde{h}_\mathbf{x}(\mathbf{y}) = \frac{1}{H_0} \sum_{l=1}^L H(\mathbf{x}, \mathbf{y}^l) \delta(\mathbf{y} - \mathbf{y}^l)$. Equation (A-4) indicates that $\mathcal{J}_{\mathrm{CDSM}}(\theta)$ can be approximately implemented based on samples using our resampling-by-compatibility strategy. More concretely, the last line in Eq. (A-4) can be implemented by sequentially dropping $\mathbf{x}$ from $p$, generating samples $\mathbf{Y}_\mathbf{x}$ to construct $\tilde{h}_\mathbf{x}$, sampling $\mathbf{y}$ from $\tilde{h}_\mathbf{x}$, and computing $\mathcal{J}_{\mathbf{x},\mathbf{y}}$ on $(\mathbf{x}, \mathbf{y})$. Note that dropping sample $\mathbf{y}$ from $\tilde{h}_\mathbf{x}$ means to choose a $\mathbf{y}$ from $\mathbf{Y}_\mathbf{x}$ with the probability proportional to $H(\mathbf{x}, \mathbf{y}^l)$.

**Training by fitting noise.**   Based on Eq. (A-1), $p_{t|0}(\mathbf{y}_t|\mathbf{y})$ is a Gaussian distribution. We denote the $\sigma_t \mathbf{I}$ as the standard variation of $p_{t|0}(\mathbf{y}_t|\mathbf{y})$, *i.e.*, $\sigma_t^2 = \frac{1}{2 \log \alpha}(\alpha^{2t} - 1)$ for VE-SDE, and $\sigma_t^2 = 1 - e^{h(t)}$ for VP-SDE. Using the reparameterization trick [2], given $\mathbf{x}, \mathbf{y}$ sampled using our resampling-by-compatibility, we have $\mathbf{y}_t = \mathbf{y} + \sigma_t \boldsymbol{\epsilon}$ for VE-SDE, and $\mathbf{y}_t = e^{\frac{1}{2}h(t)}\mathbf{y} + \sigma_t \boldsymbol{\epsilon}$ for VP-SDE, where $\boldsymbol{\epsilon} \sim \mathcal{N}(0, \mathbf{I})$. Further, $\nabla_{\mathbf{y}_t} \log p_{t|0}(\mathbf{y}_t|\mathbf{y}) = -\frac{1}{\sigma_t}\boldsymbol{\epsilon}$. Therefore, the loss $\mathcal{J}_{\mathbf{x},\mathbf{y}}$ in Eq. (A-2) can be written as

$$\mathcal{J}_{\mathbf{x},\mathbf{y}} = \mathbb{E}_{t, \boldsymbol{\epsilon} \sim \mathcal{N}(0,\mathbf{I})} \left[ \frac{w_t}{\sigma_t^2} \|s_\theta(\boldsymbol{u}_t(\mathbf{y}) + \sigma_t\boldsymbol{\epsilon}; \mathbf{x}, t)\sigma_t + \boldsymbol{\epsilon}\|_2^2 \right]. \tag{A-5}$$

where $\boldsymbol{u}_t(\mathbf{y}) = \mathbf{y}$ for VE-SDE, and $\boldsymbol{u}_t(\mathbf{y}) = e^{\frac{1}{2}h(t)}\mathbf{y}$ for VP-SDE. Equation (A-5) implies that $s_\theta(\mathbf{y}_t; \mathbf{x}, t)$ is trained to fit the scaled noise $-\frac{1}{\sigma_t}\boldsymbol{\epsilon}$.

**Pseudo-codes of algorithm training $s_\theta$.**   The pseudo-codes of the algorithm to train $s_\theta$ for the case where training data consist of condition dataset $\mathcal{D}_\mathbf{x}$ and target dataset $\mathcal{D}_\mathbf{y}$ are given in Algorithm 2. The pseudo-codes of the algorithm to train $s_\theta$ for the case with continuous distributions $p, q$ are given in Algorithm 3.

# B   Proofs

## B.1   Proof of Proposition 1

**Proposition 1.** *Let* $\mathcal{C}(\mathbf{x}, \mathbf{y}) = \frac{1}{p(\mathbf{x})}\delta(\mathbf{x} - \mathbf{x}_{\mathrm{cond}}(\mathbf{y}))$ *where* $\delta$ *is the Dirac delta function, then* $\mathcal{J}_{\mathrm{DSM}}(\theta)$ *in Eq. (1) can be reformulated as*

$$\mathcal{J}_{\mathrm{DSM}}(\theta) = \mathbb{E}_t w_t \mathbb{E}_{\mathbf{x} \sim p} \mathbb{E}_{\mathbf{y} \sim q} \mathcal{C}(\mathbf{x}, \mathbf{y}) \mathbb{E}_{\mathbf{y}_t \sim p_{t|0}(\mathbf{y}_t|\mathbf{y})} \|s_\theta(\mathbf{y}_t; \mathbf{x}, t) - \nabla_{\mathbf{y}_t} \log p_{t|0}(\mathbf{y}_t|\mathbf{y})\|_2^2. \tag{A-6}$$

*Furthermore,* $\gamma(\mathbf{x}, \mathbf{y}) = \mathcal{C}(\mathbf{x}, \mathbf{y})p(\mathbf{x})q(\mathbf{y})$ *is a joint distribution for marginal distributions* $p$ *and* $q$.

**Algorithm 2:** Training algorithm for discrete datasets

---

**Input:** Condition dataset $\mathcal{D}_{\mathbf{x}}$, target dataset $\mathcal{D}_{\mathbf{y}}$, paired data (if available)
**Output:** Trained conditional score-based model $s_{\hat{\theta}}$
Learning potentials $u_{\hat{\omega}}, v_{\hat{\omega}}$ using Algorithm 1;
`// Computing and storing` $H$ `for target samples with non-zero` $H$
$Dict = \{\}$;
**for** $\mathbf{x}$ *in* $\mathcal{D}_{\mathbf{x}}$ **do**
    $\boldsymbol{Y}_{\mathbf{x}} = \{\mathbf{y} : H(\mathbf{x}, \mathbf{y}) > 0, \mathbf{y} \in \mathcal{D}_{\mathbf{y}}\}$;
    $\boldsymbol{H}_{\mathbf{x}} = \{\frac{H(\mathbf{x}, \mathbf{y})}{H_0} : \mathbf{y} \in \boldsymbol{Y}_{\mathbf{x}}\}$, where $H_0 = \sum_{\mathbf{y} \in \boldsymbol{Y}_{\mathbf{x}}} H(\mathbf{x}, \mathbf{y})$;
    $Dict = Dict \cup \{(\boldsymbol{Y}_{\mathbf{x}}, \boldsymbol{H}_{\mathbf{x}})\}$;
**end**
`// Traning` $s_{\theta}$ `on mini-batch data`
**for** iter $= 1, \cdots, N_{\text{iter}}$ **do**
    Sampling mini-batch data $\{\mathbf{x}_b\}_{b=1}^B$ from $\mathcal{D}_{\mathbf{x}}$;
    **for** $b = 1, 2, \cdots, B$ **do**
        `// Resampling-by-compatibility`
        Finding $(\boldsymbol{Y}_{\mathbf{x}_b}, \boldsymbol{H}_{\mathbf{x}_b})$ in $Dict$ ;
        Choosing $\mathbf{y}_b$ from $\boldsymbol{Y}_{\mathbf{x}_b}$ with probability $\boldsymbol{H}_{\mathbf{x}_b}$;
        Sampling $t_b$ from $\mathcal{U}([0, T])$, and $\boldsymbol{\epsilon}_b$ from $\mathcal{N}(0, \mathbf{I})$;
    **end**
    Computing loss $\frac{1}{B} \sum_{b=1}^B \frac{w_{t_b}}{\sigma_{t_b}^2} \|s_\theta (\boldsymbol{u}_{t_b}(\mathbf{y}_b) + \sigma_{t_b}\boldsymbol{\epsilon}_b; \mathbf{x}_b, t_b) \sigma_{t_b} + \boldsymbol{\epsilon}_b\|_2^2$ ;    `//` Eq. (A-5)
    Backward propagation to compute the gradient *w.r.t.* $\theta$ and update $\theta$ using Adam algorithm;
**end**
$\hat{\theta} = \theta$.

---

**Algorithm 3:** Training algorithm for continuous distributions

---

**Input:** Condition distribution $p$, data distribution $q$, paired data (if available)
**Output:** Trained conditional score-based model $s_{\hat{\theta}}$
Learning potentials $u_{\hat{\omega}}, v_{\hat{\omega}}$ using Algorithm 1;
`// Traning` $s_{\theta}$ `on mini-batch data`
**for** iter $= 1, \cdots, N_{\text{iter}}$ **do**
    Sampling mini-batch data $\{\mathbf{x}_b\}_{b=1}^B$ from $p$;
    **for** $b = 1, 2, \cdots, B$ **do**
        `// Resampling-by-compatibility`
        Sampling $\boldsymbol{Y}_{\mathbf{x}} = \{\mathbf{y}^l\}_{l=1}^L$ from $q$;
        Computing $h^l = H(\mathbf{x}, \mathbf{y}^l)$ for all $l$ as in Eq. (7);
        Choosing $\mathbf{y}_b$ from $\boldsymbol{Y}_{\mathbf{x}}$ with probability $\frac{1}{\sum_{l=1}^L h^l}(h^1, h^2, \cdots, h^L)$;
        Sampling $t_b$ from $\mathcal{U}([0, T])$, and $\boldsymbol{\epsilon}_b$ from $\mathcal{N}(0, \mathbf{I})$;
    **end**
    Computing loss $\frac{1}{B} \sum_{b=1}^B \frac{w_{t_b}}{\sigma_{t_b}^2} \|s_\theta (\boldsymbol{u}_{t_b}(\mathbf{y}_b) + \sigma_{t_b}\boldsymbol{\epsilon}_b; \mathbf{x}_b, t_b) \sigma_{t_b} + \boldsymbol{\epsilon}_b\|_2^2$ ;    `//` Eq. (A-5)
    Backward propagation to compute the gradient *w.r.t.* $\theta$ and update $\theta$ using Adam algorithm;
**end**
$\hat{\theta} = \theta$.

---

**Proof.** We first i) prove Eq. (A-6), and then ii) show that $\gamma(\mathbf{x}, \mathbf{y})$ is a joint distribution for marginal distributions $p$ and $q$.

i) The right side of Eq. (A-6) is

$$\mathbb{E}_t w_t \mathbb{E}_{\mathbf{x} \sim p} \mathbb{E}_{\mathbf{y} \sim q} \mathcal{C}(\mathbf{x}, \mathbf{y}) \mathbb{E}_{\mathbf{y}_t \sim p_{t|0}(\mathbf{y}_t|\mathbf{y})} \left\| s_\theta(\mathbf{y}_t; \mathbf{x}, t) - \nabla_{\mathbf{y}_t} \log p_{t|0}(\mathbf{y}_t|\mathbf{y}) \right\|_2^2$$

$$= \mathbb{E}_t w_t \mathbb{E}_{\mathbf{y} \sim q} \int p(\mathbf{x}) \mathcal{C}(\mathbf{x}, \mathbf{y}) \mathbb{E}_{\mathbf{y}_t \sim p_{t|0}(\mathbf{y}_t|\mathbf{y})} \left\| s_\theta(\mathbf{y}_t; \mathbf{x}, t) - \nabla_{\mathbf{y}_t} \log p_{t|0}(\mathbf{y}_t|\mathbf{y}) \right\|_2^2 \mathrm{d}\mathbf{x}$$

$$=\mathbb{E}_t w_t \mathbb{E}_{\mathbf{y} \sim q} \int \delta(\mathbf{x} - \mathbf{x}_{\mathrm{cond}}(\mathbf{y})) \mathbb{E}_{\mathbf{y}_t \sim p_{t|0}(\mathbf{y}_t|\mathbf{y})} \left\| s_\theta(\mathbf{y}_t; \mathbf{x}, t) - \nabla_{\mathbf{y}_t} \log p_{t|0}(\mathbf{y}_t|\mathbf{y}) \right\|_2^2 \mathrm{d}\mathbf{x}$$

$$=\mathbb{E}_t w_t \mathbb{E}_{\mathbf{y} \sim q} \mathbb{E}_{\mathbf{y}_t \sim p_{t|0}(\mathbf{y}_t|\mathbf{y})} \left\| s_\theta(\mathbf{y}_t; \mathbf{x}_{\mathrm{cond}}(\mathbf{y}), t) - \nabla_{\mathbf{y}_t} \log p_{t|0}(\mathbf{y}_t|\mathbf{y}) \right\|_2^2,$$

which is the definition of $\mathcal{J}_{\mathrm{DSM}}(\theta)$ in Eq. (1).

ii) We show that the marginal distributions of $\gamma(\mathbf{x}, \mathbf{y})$ are respectively $p$ and $q$ as follows. Firstly,

$$\int \gamma(\mathbf{x}, \mathbf{y}) \, \mathrm{d}\mathbf{x} = \int \delta(\mathbf{x} - \mathbf{x}_{\mathrm{cond}}(\mathbf{y})) q(\mathbf{y}) \, \mathrm{d}\mathbf{x} = q(\mathbf{y}) \int \delta(\mathbf{x} - \mathbf{x}_{\mathrm{cond}}(\mathbf{y})) \, \mathrm{d}\mathbf{x} = q(\mathbf{y}). \quad \text{(A-7)}$$

Secondly, from the definition of $\delta(\cdot)$, we have $\delta(\mathbf{x} - \mathbf{x}_{\mathrm{cond}}(\mathbf{y})) = \sum_{\{\mathbf{y}': \mathbf{x}_{\mathrm{cond}}(\mathbf{y}')=\mathbf{x}\}} \delta(\mathbf{y} - \mathbf{y}')$. Then, we have

$$\int \gamma(\mathbf{x}, \mathbf{y}) \, \mathrm{d}\mathbf{y} = \int \delta(\mathbf{x} - \mathbf{x}_{\mathrm{cond}}(\mathbf{y})) q(\mathbf{y}) \, \mathrm{d}\mathbf{y}$$

$$= \int \sum_{\{\mathbf{y}': \mathbf{x}_{\mathrm{cond}}(\mathbf{y}')=\mathbf{x}\}} \delta(\mathbf{y}' - \mathbf{y}) q(\mathbf{y}) \, \mathrm{d}\mathbf{y}$$

$$= \sum_{\{\mathbf{y}': \mathbf{x}_{\mathrm{cond}}(\mathbf{y}')=\mathbf{x}\}} \int \delta(\mathbf{y}' - \mathbf{y}) q(\mathbf{y}) \, \mathrm{d}\mathbf{y} \qquad \text{(A-8)}$$

$$= \sum_{\{\mathbf{y}': \mathbf{x}_{\mathrm{cond}}(\mathbf{y}')=\mathbf{x}\}} q(\mathbf{y}')$$

$$= p(\mathbf{x}).$$

## B.2 Proof of Theorem 1

**Theorem 1.** *For $\mathbf{x} \sim p$, we define the forward SDE $\mathrm{d}\mathbf{y}_t = f(\mathbf{y}_t, t) \, \mathrm{d}t + g(t) \, \mathrm{d}\mathbf{w}$ with $\mathbf{y}_0 \sim \hat{\pi}(\cdot|\mathbf{x})$ and $t \in [0, T]$, where $f, g, T$ are given in Appendix A.1. Let $p_t(\mathbf{y}_t|\mathbf{x})$ be the corresponding distribution of $\mathbf{y}_t$ and $\mathcal{J}_{\mathrm{CSM}}(\theta) = \mathbb{E}_t w_t \mathbb{E}_{\mathbf{x} \sim p} \mathbb{E}_{\mathbf{y}_t \sim p_t(\mathbf{y}_t|\mathbf{x})} \| s_\theta(\mathbf{y}_t; \mathbf{x}, t) - \nabla_{\mathbf{y}_t} \log p_t(\mathbf{y}_t|\mathbf{x}) \|_2^2$, then we have $\nabla_\theta \mathcal{J}_{\mathrm{CDSM}}(\theta) = \nabla_\theta \mathcal{J}_{\mathrm{CSM}}(\theta)$.*

**Proof.** Given $\mathbf{x}$ and $t$, $p_t(\mathbf{y}_t|\mathbf{x})$ is the distribution of $\mathbf{y}_t$ produced by the forward SDE $\mathrm{d}\mathbf{y}_t = f(\mathbf{y}_t, t) \, \mathrm{d}t + g(t) \, \mathrm{d}\mathbf{w}$ with initial state $\mathbf{y}_0 \sim \hat{\pi}(\mathbf{y}_0|\mathbf{x})$. This implies that $\mathbf{x} \to \mathbf{y}_0 \to \mathbf{y}_t$ is a Markov Chain. So the distribution $p_{t|0}(\mathbf{y}_t|\mathbf{y}_0, \mathbf{x})$ of $\mathbf{y}_t$ given $\mathbf{y}_0$ and $\mathbf{x}$ depends on $\mathbf{y}_0$ but $\mathbf{x}$, i.e., $p_{t|0}(\mathbf{y}_t|\mathbf{y}_0, \mathbf{x}) = p_{t|0}(\mathbf{y}_t|\mathbf{y}_0)$, where $p_{t|0}(\mathbf{y}_t|\mathbf{y}_0)$ is the distribution of $\mathbf{y}_t$ by the forward SDE $\mathrm{d}\mathbf{y}_t = f(\mathbf{y}_t, t) \, \mathrm{d}t + g(t) \, \mathrm{d}\mathbf{w}$ with initial state $\mathbf{y}_0$. According to [3], given any $\mathbf{x}$ and $t$, we have

$$\mathbb{E}_{\mathbf{y}_0 \sim \hat{\pi}(\mathbf{y}_0|\mathbf{x})} \mathbb{E}_{\mathbf{y}_t \sim p_{t|0}(\mathbf{y}_t|\mathbf{y}_0, \mathbf{x})} \| s_\theta(\mathbf{y}_t; \mathbf{x}, t) - \nabla_{\mathbf{y}_t} \log p_{t|0}(\mathbf{y}_t|\mathbf{y}_0, \mathbf{x}) \|_2^2$$

$$= \mathbb{E}_{\mathbf{y}_t \sim p_t(\mathbf{y}_t|\mathbf{x})} \| s_\theta(\mathbf{y}_t; \mathbf{x}, t) - \nabla_{\mathbf{y}_t} \log p_t(\mathbf{y}_t|\mathbf{x}) \|_2^2 + C_{\mathbf{x}, t}, \qquad \text{(A-9)}$$

where $C_{\mathbf{x}, t}$ is a constant to $\theta$ depending on $\mathbf{x}$ and $t$. Then, we have

$$w_t \left( \mathbb{E}_{\mathbf{y}_t \sim p_t(\mathbf{y}_t|\mathbf{x})} \| s_\theta(\mathbf{y}_t; \mathbf{x}, t) - \nabla_{\mathbf{y}_t} \log p_t(\mathbf{y}_t|\mathbf{x}) \|_2^2 + C_{\mathbf{x}, t} \right)$$

$$= w_t \mathbb{E}_{\mathbf{y}_0 \sim \hat{\pi}(\mathbf{y}_0|\mathbf{x})} \mathbb{E}_{\mathbf{y}_t \sim p_{t|0}(\mathbf{y}_t|\mathbf{y}_0)} \| s_\theta(\mathbf{y}_t; \mathbf{x}, t) - \nabla_{\mathbf{y}_t} \log p_{t|0}(\mathbf{y}_t|\mathbf{y}_0) \|_2^2. \qquad \text{(A-10)}$$

Taken expectation over $\mathbf{x}$ and $t$ in the above equation, we have

$$\mathcal{J}_{\mathrm{CSM}}(\theta) + \mathbb{E}_{\mathbf{x} \sim p} \mathbb{E}_t w_t C_{\mathbf{x}, t}$$

$$= \mathbb{E}_t w_t \mathbb{E}_{\mathbf{x} \sim p} \mathbb{E}_{\mathbf{y}_0 \sim \hat{\pi}(\mathbf{y}_0|\mathbf{x})} \mathbb{E}_{\mathbf{y}_t \sim p_{t|0}(\mathbf{y}_t|\mathbf{y}_0)} \| s_\theta(\mathbf{y}_t; \mathbf{x}, t) - \nabla_{\mathbf{y}_t} \log p_{t|0}(\mathbf{y}_t|\mathbf{y}_0) \|_2^2$$

$$= \mathbb{E}_t w_t \mathbb{E}_{\mathbf{x} \sim p} \mathbb{E}_{\mathbf{y}_0 \sim q} H(\mathbf{x}, \mathbf{y}_0) \mathbb{E}_{\mathbf{y}_t \sim p_{t|0}(\mathbf{y}_t|\mathbf{y}_0)} \| s_\theta(\mathbf{y}_t; \mathbf{x}, t) - \nabla_{\mathbf{y}_t} \log p_{t|0}(\mathbf{y}_t|\mathbf{y}_0) \|_2^2 \qquad \text{(A-11)}$$

$$= \mathcal{J}_{\mathrm{CDSM}}(\theta).$$

Since $\mathbb{E}_{\mathbf{x} \sim p} \mathbb{E}_t w_t C_{\mathbf{x}, t}$ is a constant to $\theta$, we have

$$\nabla_\theta \mathcal{J}_{\mathrm{CDSM}}(\theta) = \nabla_\theta \mathcal{J}_{\mathrm{CSM}}(\theta). \qquad \text{(A-12)}$$

## B.3 Assumptions and Proof of Theorem 2

**Theorem 2.** *Suppose the assumptions in Appendix B.3.1 hold, and $w_t = g(t)^2$, then we have*

$$\mathbb{E}_{\mathbf{x}\sim p}W_2\left(p^{\text{sde}}(\cdot|\mathbf{x}), \pi(\cdot|\mathbf{x})\right) \leq C_1\left\|\nabla_{\hat{\pi}}\mathcal{L}(\hat{\pi}, u_{\hat{\omega}}, v_{\hat{\omega}})\right\|_1 + \sqrt{C_2\mathcal{J}_{\text{CSM}}(\hat{\theta})}$$
$$+ C_3\mathbb{E}_{\mathbf{x}\sim p}W_2\left(p_T(\cdot|\mathbf{x}), p_{\text{prior}}\right), \tag{A-13}$$

*where $C_1, C_2,$ and $C_3$ are constants to $\hat{\omega}$ and $\hat{\theta}$.*

### B.3.1 Assumptions

(1) $f(\mathbf{y}, t)$ is Lipschitz continuous in the space variable $\mathbf{y}$: there exists a positive constant $L_f(t) \in (0, \infty)$ depending on $t \in [0, T]$ such that for all $\mathbf{y}_1, \mathbf{y}_2$,

$$\|f(\mathbf{y}_1, t) - f(\mathbf{y}_2, t)\|_2 \leq L_f(t)\|\mathbf{y}_1 - \mathbf{y}_2\|_2. \tag{A-14}$$

(2) $s_\theta(\mathbf{y}; \mathbf{x}, t)$ satisfies the one-sided Lipschitz condition: there exists a constant $L_s(t)$ depending on $t$, such that for all $\mathbf{y}_1, \mathbf{y}_2$,

$$(s_\theta(\mathbf{y}_1; \mathbf{x}, t) - s_\theta(\mathbf{y}_2; \mathbf{x}, t))(\mathbf{y}_1 - \mathbf{y}_2) \leq L_s(t)\|\mathbf{y}_1 - \mathbf{y}_2\|_2, \tag{A-15}$$

for any $\mathbf{x}$.

(3) For any $\mathbf{x}$, $\mathbb{E}_{\hat{\pi}(\cdot|\mathbf{x})}[|\log \hat{\pi}(\cdot|\mathbf{x})|], \mathbb{E}_{\hat{\pi}(\cdot|\mathbf{x})}[\log|\Lambda(\mathbf{y})|], \mathbb{E}_{p_{\text{prior}}}[|\log p_{\text{prior}}|]$, and $\mathbb{E}_{p_{\text{prior}}}[\log|\Lambda(\mathbf{y})|]$ are finite, where $\Lambda(\mathbf{y}) = \log\max(\|\mathbf{y}\|_2, 1)$.

(4) There exists positive constants $A_1$ and $A_2$ such that

$$f(\mathbf{y}, t)\mathbf{y} \leq A_1\|\mathbf{y}\|_2 + A_2, \forall \mathbf{y}, \forall t \in [0, T]. \tag{A-16}$$

(5) There exists a positive constant $A_3$ such that

$$\frac{1}{A_3} < g(t) < A_3, \forall t \in [0, T]. \tag{A-17}$$

(6) $\int_0^T \mathbb{E}_{p_t(\cdot|\mathbf{x})}[f^2]\,\mathrm{d}t, \int_0^T \mathbb{E}_{q_t(\cdot|\mathbf{x})}[(f - g^2 s_\theta)]\,\mathrm{d}t$ are finite for any $\mathbf{x}$, where $q_t(\mathbf{y}_t|\mathbf{x})$ is the distribution produced by the reverse SDE in Eq. (10) at time $t$.

(7) $\hat{\pi}(\cdot|\mathbf{x}), p_{\text{prior}}$ are in $C^2$ *w.r.t.* $\mathbf{y}$ for any $\mathbf{x}$. $f, g, s_\theta$ are in $C^2$ *w.r.t.* $\mathbf{y}$ and $t$ for any $\mathbf{x}$.

(8) There exists $k > 0$ such that $p_t(\mathbf{y}|\mathbf{x}) = \mathcal{O}(\exp(-\|\mathbf{y}\|_2^k))$ and $q_t(\mathbf{y}|\mathbf{x}) = \mathcal{O}(\exp(-\|\mathbf{y}\|_2^k))$ for any $t \in [0, T]$ and any $\mathbf{x}$.

(9) $\mathcal{L}(\pi, u, v)$ is $\kappa$-strongly convex in $L_1$-norm *w.r.t.* $\pi$.

Assumptions (1)-(8) are based on the assumptions in [4] that investigates the bound for unconditional SBDMs. For Assumption (9), $\mathcal{L}(\pi, u, v)$ is strongly convex as proved in [5].

### B.3.2 Proof

Since $W_2(\cdot, \cdot)$ is a proper metric, using the triangle inequality, we have

$$\mathbb{E}_{\mathbf{x}\sim p}W_2\left(p^{\text{sde}}(\cdot|\mathbf{x}), \pi(\cdot|\mathbf{x})\right) \leq \mathbb{E}_{\mathbf{x}\sim p}W_2\left(p^{\text{sde}}(\cdot|\mathbf{x}), \hat{\pi}(\cdot|\mathbf{x})\right) + \mathbb{E}_{\mathbf{x}\sim p}W_2\left(\hat{\pi}(\cdot|\mathbf{x}), \pi(\cdot|\mathbf{x})\right). \tag{A-18}$$

We next respectively bound the right-side terms.

**Bounding $\mathbb{E}_{\mathbf{x}\sim p}W_2\left(p^{\text{sde}}(\cdot|\mathbf{x}), \hat{\pi}(\cdot|\mathbf{x})\right)$.** Let $I(t) = \exp\left(\int_0^t L_f(r) + L_s(r)g(r)^2\,\mathrm{d}r\right)$. According to Corollary 1 in [4], for any $\mathbf{x}$, we have

$$W_2\left(p^{\text{sde}}(\cdot|\mathbf{x}), \hat{\pi}(\cdot|\mathbf{x})\right) \leq \sqrt{T\left(\int_0^T g(t)^2 I(t)^2\,\mathrm{d}t\right)\mathcal{J}_{\text{SM}}^{\mathbf{x}}(\hat{\theta})} + I(T)W_2\left(p_T(\cdot|\mathbf{x}), p_{\text{prior}}\right), \tag{A-19}$$

where $\mathcal{J}_{\mathrm{SM}}^{\mathbf{x}}(\hat{\theta}) = \mathbb{E}_t w_t \mathbb{E}_{\mathbf{y}_t \sim p_t(\mathbf{y}|\mathbf{x})} \| s_{\hat{\theta}}(\mathbf{y}_t; \mathbf{x}, t) - \nabla_{\mathbf{y}_t} \log p_t(\mathbf{y}_t|\mathbf{x}) \|_2^2$. Taking expectation over $\mathbf{x}$ in Eq. (A-19), we have

$$
\mathbb{E}_{\mathbf{x} \sim p} W_2 \left( p^{\mathrm{sde}}(\cdot|\mathbf{x}), \hat{\pi}(\cdot|\mathbf{x}) \right) \leq \mathbb{E}_{\mathbf{x} \sim p} \left( \sqrt{T \left( \int_0^T g(t)^2 I(t)^2 \, \mathrm{d}t \right) \mathcal{J}_{\mathrm{SM}}^{\mathbf{x}}(\hat{\theta})} \right) \tag{A-20}
$$
$$
+ I(T) \mathbb{E}_{\mathbf{x} \sim p} W_2 \left( p_T(\cdot|\mathbf{x}), p_{\mathrm{prior}} \right).
$$

Since $\sqrt{x}$ is concave in $[0, \infty)$, using the Jesen-Inequality, we have $\mathbb{E}[\sqrt{x}] \leq \sqrt{\mathbb{E}[x]}$. Therefore,

$$
\mathbb{E}_{\mathbf{x} \sim p} W_2 \left( p^{\mathrm{sde}}(\cdot|\mathbf{x}), \hat{\pi}(\cdot|\mathbf{x}) \right)
$$
$$
\leq \sqrt{T \left( \int_0^T g(t)^2 I(t)^2 \, \mathrm{d}t \right) \mathbb{E}_{\mathbf{x} \sim p}[\mathcal{J}_{\mathrm{SM}}^{\mathbf{x}}(\hat{\theta})]} + I(T) \mathbb{E}_{\mathbf{x} \sim p} W_2 \left( p_T(\cdot|\mathbf{x}), p_{\mathrm{prior}} \right) \tag{A-21}
$$
$$
= \sqrt{T \left( \int_0^T g(t)^2 I(t)^2 \, \mathrm{d}t \right) \mathcal{J}_{\mathrm{CSM}}(\hat{\theta})} + I(T) \mathbb{E}_{\mathbf{x} \sim p} W_2 \left( p_T(\cdot|\mathbf{x}), p_{\mathrm{prior}} \right).
$$

Let $C_2 = T \left( \int_0^T g(t)^2 I(t)^2 \, \mathrm{d}t \right)$ and $C_3 = I(T)$. Then, we have

$$
\mathbb{E}_{\mathbf{x} \sim p} W_2 \left( p^{\mathrm{sde}}(\cdot|\mathbf{x}), \hat{\pi}(\cdot|\mathbf{x}) \right) \leq \sqrt{C_2 \mathcal{J}_{\mathrm{CSM}}(\hat{\theta})} + C_3 \mathbb{E}_{\mathbf{x} \sim p} W_2 \left( p_T(\cdot|\mathbf{x}), p_{\mathrm{prior}} \right). \tag{A-22}
$$

**Bounding $\mathbb{E}_{\mathbf{x} \sim p} W_2 \left( \hat{\pi}(\cdot|\mathbf{x}), \pi(\cdot|\mathbf{x}) \right)$.** According to Remark 2.26 in [6] (the relation between the Wasserstein distance and $L_1$-distance), we have

$$
W_2(\mu, \nu) \leq \max_{\mathbf{x}, \mathbf{y} \in \mathcal{X}} \{ \|\mathbf{x} - \mathbf{y}\|_2 \} \|\mu - \nu\|_1, \tag{A-23}
$$

for any $\mu, \nu$ supported on $\mathcal{X}$. We then have

$$
W_2 \left( \hat{\pi}(\cdot|\mathbf{x}), \pi(\cdot|\mathbf{x}) \right) \leq \max_{\mathbf{y}, \mathbf{y}' \in \mathcal{Y}} \{ \|\mathbf{y} - \mathbf{y}'\|_2 \} \|\hat{\pi}(\cdot|\mathbf{x}) - \pi(\cdot|\mathbf{x})\|_1 = \eta \|\hat{\pi}(\cdot|\mathbf{x}) - \pi(\cdot|\mathbf{x})\|_1 \tag{A-24}
$$

for any $\mathbf{x}$, where we denote $\eta = \max_{\mathbf{y}, \mathbf{y}' \in \mathcal{Y}} \{ \|\mathbf{y} - \mathbf{y}'\|_2 \}$. Therefore,

$$
\mathbb{E}_{\mathbf{x} \sim p} W_2 \left( \hat{\pi}(\cdot|\mathbf{x}), \pi(\cdot|\mathbf{x}) \right) \leq \eta \mathbb{E}_{\mathbf{x} \sim p} \|\hat{\pi}(\cdot|\mathbf{x}) - \pi(\cdot|\mathbf{x})\|_1
$$
$$
= \eta \int p(\mathbf{x}) \int \left| \hat{\pi}(\mathbf{y}|\mathbf{x}) - \pi(\mathbf{y}|\mathbf{x}) \right| \, \mathrm{d}\mathbf{y} \, \mathrm{d}\mathbf{x}
$$
$$
= \eta \int \left| p(\mathbf{x}) \hat{\pi}(\mathbf{y}|\mathbf{x}) - p(\mathbf{x}) \pi(\mathbf{y}|\mathbf{x}) \right| \, \mathrm{d}\mathbf{y} \, \mathrm{d}\mathbf{x} \tag{A-25}
$$
$$
= \eta \int \left| \hat{\pi}(\mathbf{x}, \mathbf{y}) - \pi(\mathbf{x}, \mathbf{y}) \right| \, \mathrm{d}\mathbf{x} \, \mathrm{d}\mathbf{y}
$$
$$
= \eta \|\hat{\pi} - \pi\|_1.
$$

By virtue to Theorem 4.3 in [5], we have

$$
\|\hat{\pi} - \pi\|_1 \leq \frac{1}{\kappa} \left\| \nabla_{\hat{\pi}} \mathcal{L}(\hat{\pi}, u_{\hat{\omega}}, v_{\hat{\omega}}) \right\|_1. \tag{A-26}
$$

We therefore have

$$
\mathbb{E}_{\mathbf{x} \sim p} W_2(\hat{\pi}(\cdot|\mathbf{x}), \pi(\cdot|\mathbf{x})) \leq \frac{\eta}{\kappa} \left\| \nabla_{\hat{\pi}} \mathcal{L}(\hat{\pi}, u_{\hat{\omega}}, v_{\hat{\omega}}) \right\|_1. \tag{A-27}
$$

Let $C_1 = \frac{\eta}{\kappa}$, we have

$$
\mathbb{E}_{\mathbf{x} \sim p} W_2(\hat{\pi}(\cdot|\mathbf{x}), \pi(\cdot|\mathbf{x})) \leq C_1 \left\| \nabla_{\hat{\pi}} \mathcal{L}(\hat{\pi}, u_{\hat{\omega}}, v_{\hat{\omega}}) \right\|_1. \tag{A-28}
$$

Combining Eqs. (A-18), (A-22), and (A-28), we have

$$
\mathbb{E}_{\mathbf{x} \sim p} W_2 \left( p^{\mathrm{sde}}(\cdot|\mathbf{x}), \pi(\cdot|\mathbf{x}) \right) \leq C_1 \left\| \nabla_{\hat{\pi}} \mathcal{L}(\hat{\pi}, u_{\hat{\omega}}, v_{\hat{\omega}}) \right\|_1 + \sqrt{C_2 \mathcal{J}_{\mathrm{CSM}}(\hat{\theta})} \tag{A-29}
$$
$$
+ C_3 \mathbb{E}_{\mathbf{x} \sim p} W_2 \left( p_T(\cdot|\mathbf{x}), p_{\mathrm{prior}} \right).
$$

The proof is completed.

## C Experimental Details

We provide the details for learning the potentials $u_\omega(\mathbf{x}), v_\omega(\mathbf{y})$, training the conditional score-based model $s_\theta(\mathbf{y}; \mathbf{x}, t)$, generating data in inference, and computing the metric Acc. All the experiments are conducted using 2 NVIDIA Tesla V100 32GB GPUs. The codes are in pytorch [7].

### C.1 Details for Toy Data Experiment in Figure 2

**Architectures of $u_\omega, v_\omega$.** The architectures of both of $u_\omega$ and $v_\omega$ are FC(1,1024) $\rightarrow$ Tanh $\rightarrow$ FC(1024,1), where FC($a, b$) is the fully-connected layer with input/output dimension of $a/b$ and Tanh is the activation function.

**Details for learning $u_\omega, v_\omega$.** We use the $L_2$-regularized unsupervised OT where $c$ is taken as the squared $L_2$-distance. The learning rate is 1e-5. The batch size $B'$ is set to 256. The Adam algorithm is employed to update the parameters.

**Architecture of $s_\theta$.** The backbone of $s_\theta$ is FC(1,512) $\rightarrow$ SiLU $\rightarrow$ FC(512,512) $\rightarrow$ SiLU $\rightarrow$ FC(512,1), where SiLU is the activation function. We add the embedding of time $t$ and condition $\mathbf{x}$ to the activation of SiLU. The embedding block for $t$ is GaussianFourierProjection(256) $\rightarrow$ FC(256,512) $\rightarrow$ SiLU $\rightarrow$ FC(512,512). The embedding block for $\mathbf{x}$ is FC(1,512) $\rightarrow$ SiLU $\rightarrow$ FC(512,512) $\rightarrow$ SiLU $\rightarrow$ FC(512,512). The GaussianFourierProjection has been adopted in [1].

**Details for training $s_\theta$ and inference.** We take the VE-SDE with $\alpha = 25$, and $T = 1$. We set $w_t = \sigma_t^2$, the batch size $B = 32$, $L = 10B$ in Algorithm 3. The learning rate is 1e-4. The Adam algorithm and the exponential moving average for model parameters with decay=0.999 are applied. We take the Euler-Maruyama method to perform the reverse SDE for generating data in inference. The initial state $\mathbf{y}_T$ is sampled from the $p_{\mathrm{prior}} = \mathcal{N}(0, \sigma_T^2 \mathbf{I})$.

### C.2 Details for Unpaired Super-Resolution

**Architectures of $u_\omega, v_\omega$.** The architectures of $u_\omega$ and $v_\omega$ are FC(12288,512) $\rightarrow$ SiLU $\rightarrow$ FC(512,512) $\rightarrow$ SiLU $\rightarrow$ FC(512,512) $\rightarrow$ SiLU $\rightarrow$ FC(512,512) $\rightarrow$ SiLU $\rightarrow$ FC(512,512) $\rightarrow$ SiLU $\rightarrow$ FC(512,512) $\rightarrow$ SiLU $\rightarrow$ FC(512,1). We reshape the input images from size (64,64,3) to size 12288. The design of the architectures is inspired by [5].

**Details for learning $u_\omega, v_\omega$.** We use the $L_2$-regularized unsupervised OT where $c$ is taken as the mean squared $L_2$-distance (following [5]), and $\epsilon$ is set to 1e-7. The learning rate is 1e-6. The batch size $B'$ is set to 64. The Adam algorithm is employed to update the parameters.

**Architecture of $s_\theta$.** The backbone of $s_\theta$ is based on the architecture of DDIM [8] on CelebA dataset for unconditional image generation. We apply the condition to the backbone by concatenating the degenerated image $\mathbf{x}$ with the noisy image $\mathbf{y}_t$ as input, inspired by [9] that tackles the paired super-resolution.

**Details for training $s_\theta$ and inference.** We take the VP-SDE with $\beta_{\min} = 0.1, \beta_{\max} = 20$, and $T = 1$. We set $w_t = \sigma_t^2$, the batch size $B = 64$ in Algorithm 2. The learning rate is $2e - 4$. The Adam algorithm and the exponential moving average for model parameters with decay=0.999 are applied. To facilitate the training, we take the trained model in [8] on CelebA images as initialization. In inference, we take the sampling method in DDIM to perform the reverse SDE to generate data. Following [10, 11], we add noise to the low-resolution images by sampling $\mathbf{y}_M$ from $p_{M|0}(\mathbf{y}_M|\mathbf{x})$ as the initial state. $M$ is set to 0.2.

### C.3 Details for Semi-paired Image-to-Image Translation on Animal Images

In experiments, we randomly choose 1000/150 images for each species for training/testing.

**Architectures of $u_\omega, v_\omega$.** The architectures of $u_\omega$ and $v_\omega$ consist of a feature extractor and a head. We take the image encoder "ViT-B/32" of CLIP [12] as the feature extractor. The feature extractor is fixed in training. The architecture of the head is the same as that of $u_\omega, v_\omega$ for unpaired super-resolution except that the input dimension is 512.

**Details for learning $u_\omega, v_\omega$.** We use the $L_2$-regularized semi-supervised OT where $c$ is taken as the cosine distance of extracted features by the above feature extractor, and $\epsilon$ is set to 1e-5. The

learning rate is 1e-6. The batch size $B'$ is set to 64. The Adam algorithm is employed to update the parameters.

**Architecture of $s_\theta$.** The architecture of $s_\theta$ is based on the architecture of model of ILVR [13] on dog images for unconditional image generation. We add the embedding of condition $\mathbf{x}$ to the output of each residual block. The embedding block for condition $\mathbf{x}$ comprises the feature extractor as mentioned above followed by an embedding module. The architecture of the embedding module is FC(512,512) $\rightarrow$ SiLU $\rightarrow$ FC(512,512).

**Details for training $s_\theta$, inference, and computing the Acc.** We take the VP-SDE with $\beta_{\min} = 0.1, \beta_{\max} = 20$, and $T = 1$. We set $w_t = \sigma_t^2$, the batch size $B = 16$ in Algorithm 2. The learning rate is $2e - 5$. The Adam algorithm and the exponential moving average for model parameters with decay=0.999 are applied. To facilitate the training, we take the trained model in [13] on dog images as initialization. In inference, we take the sampling method in DDIM to perform the reverse SDE to generate data. The initial state $\mathbf{y}_T$ is sampled from the $p_{\text{prior}} = \mathcal{N}(0, \mathbf{I})$. To compute the metric Acc, we classify the translated images using CLIP ("ViT-B/32") into the candidate classes of lion, tiger, and wolf. We then compute the precision against the ground-truth translated classes.

### C.4 Details for Semi-paired Image-to-Image Translation on Digits

**Architectures of $u_\omega, v_\omega$.** The architectures of $u_\omega$ and $v_\omega$ are the same as the architectures of $u_\omega, v_\omega$ for unpaired super-resolution except that the input dimension is 784. We reshape the input images from size (28,28) to size 784.

**Details for learning $u_\omega, v_\omega$.** We use the $L_2$-regularized semi-supervised OT where $c$ is taken as the cosine distance of extracted features by a pre-trained feature extractor, and $\epsilon$ is set to 1e-5. The learning rate is 1e-6. The batch size $B'$ is set to 64. The Adam algorithm is employed to update the parameters. We train auto-encoders (consisting of an encoder and a decoder) for MNIST and Chinese-MNIST respectively, and the encoder is taken as the feature extractor. The architecture of the encoder is Conv(1,64,4,2,0) $\rightarrow$ BN $\rightarrow$ SiLU $\rightarrow$ Conv(64,128,4,2,0) $\rightarrow$ BN $\rightarrow$ SiLU $\rightarrow$ Conv(128, 128,3,1,0) $\rightarrow$ BN $\rightarrow$ SiLU, where "BN" is the Batch Normalization layer. The architecture of the encoder is Conv(128,128,3,1,0) $\rightarrow$ BN $\rightarrow$ SiLU $\rightarrow$ Tconv(128,64,4,2,0) $\rightarrow$ BN $\rightarrow$ SiLU $\rightarrow$ Tconv(64,1,4,2,0) $\rightarrow$ Sigmoid, where Tconv is the transposed convolutional layer, and Sigmoid is the activation function. We use Adam algorithm to train the auto-encoder with learning rate 1e-4.

**Architecture of $s_\theta$.** The backbone of $s_\theta$ is the architecture of model [1] on MNIST for unconditional image generation. We add the embedding of condition $\mathbf{x}$ to the output of each residual block. The embedding block for condition $\mathbf{x}$ is FC(784,512) $\rightarrow$ SiLU $\rightarrow$ FC(512,512) $\rightarrow$ SiLU $\rightarrow$ FC(512,256).

**Details for training $s_\theta$, inference, and computing the Acc.** We take the VE-SDE with $\alpha = 25$, and $T = 1$. We set $w_t = \sigma_t^2$, the batch size $B = 32$ in Algorithm 2. The learning rate is 1e-4. The Adam algorithm and the exponential moving average for model parameters with decay=0.999 are applied. In inference, we take the Predictor-Corrector algorithm in [1] to perform the reverse SDE to generate data, where the predictor is taken as the Euler-Maruyama method. The initial state $\mathbf{y}_T$ is sampled from the $p_{\text{prior}} = \mathcal{N}(0, \sigma_T^2 \mathbf{I})$. To compute the metric Acc, we classify the translated images using a classifier (LeNet) trained on Chinese-MNIST. We then compute the precision against the ground-truth translated classes.

## D Additional Experimental Analysis and Results

### D.1 Additional Experimental Analysis

**Guided images sampled based on OT.** We show the examples of guided high-resolution images sampled based on OT in Fig. A-1. We can observe that the guided high-resolution images share similar structures to the given degenerated image.

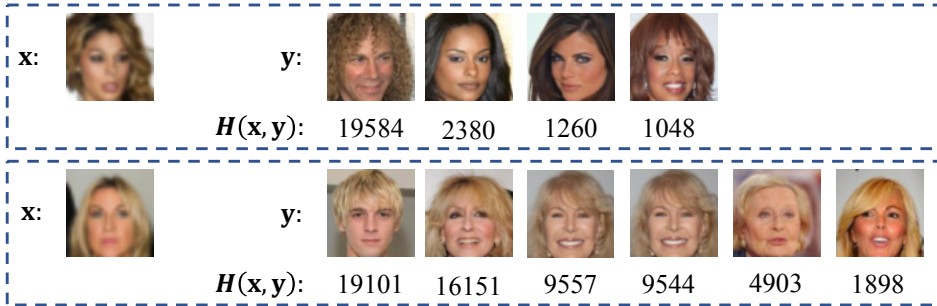

Figure A-1: Examples of guided high-resolution images (*i.e.*, $H(\mathbf{x}, \mathbf{y}) > 0$) chosen from B1 based on OT for the given degenerated low-resolution image $\mathbf{x}$ from A0 in training. Note that considering the numerical issue, we choose the guided high-resolution images $\mathbf{y}$ such that $H(\mathbf{x}, \mathbf{y}) > 0.01$.

**What happens to the compatibility on the source data with no good target to be paired?** To figure out what happens to the compatibility function $H$ when there is no good target data, we conduct the following experiments. Firstly, we count the number of source samples satisfying that there is no target sample such that $H > 0.001$, in CelebA dataset. We find that 25.9% of source samples meets such condition. Note that the other source samples are often with $H$ larger than 1000 on some target samples (since the $\epsilon$ is 1e-7 in Eq. (7)). Secondly, we add noisy images to the source dataset to train the potentials $u_\omega$ and $v_\omega$, and count the ratio of noisy images satisfying $H < 0.001$ on all target samples. The results are reported in Table A-1. The noisy images are generated from the standard normal distribution and with the same shape as the source images.

Table A-1: Ratio of noisy images with $H < 0.001$ when adding varying numbers of noisy images to the source dataset.

| Number of noisy images : Number of clean images | 0.1 : 1 | 0.2 : 1 | 0.3 : 1 | 0.4 : 1 | 0.5 : 1 |
|---|---|---|---|---|---|
| Ratio of noisy images assigned with $H < 0.001$ | 89.3% | 85.6% | 83.9% | 81.6% | 80.2% |

It can be seen that more than 80% of noisy images that are with no good target data are assigned with near-to-zero $H$ ($H < 0.001$), when the ratio of numbers of noisy images to clean images is in [0.1,0.5].

**Empirical comparison of the "soft" and "hard" coupling relationship.** To study how sparse $H$ is, for each target image $\mathbf{y}$, we denote the number of source image $\mathbf{x}$ with "non-zero $H$" as $n_\mathbf{y}$ (i.e., $n_\mathbf{y} = |\{\mathbf{x} : H(\mathbf{x}, \mathbf{y}) > 0.001\}|$, considering numerical issues) in CelebA dataset. The histogram of $n_\mathbf{y}$ is shown in the Table A-2.

Table A-2: Histogram of number $n_\mathbf{y}$ of source images with "non-zero $H$" for target image $\mathbf{y}$, where the total numbers of both source images and target images are 80k.

| Bins for $n_\mathbf{y}$ | [0,10) | [10,20) | [20,50) | [50,100) | [100,600) | [600,80k] |
|---|---|---|---|---|---|---|
| Frequency | 59600 | 8064 | 8468 | 3537 | 1716 | 0 |

We can see from Table A-2 that all the target images are with $n_\mathbf{y} \leq 600$, and more than 70% of target images are with $n_\mathbf{y} \leq 10$. This implies that for each $\mathbf{y}$ in more than 70% target images, there are no more than 10 among 80K source images $\mathbf{x}$ satisfying $H(\mathbf{x}, \mathbf{y}) > 0.001$. So $H$ is sparse to some extent. We also count the number of target images with $n_\mathbf{y} = 1$ ($n_\mathbf{y} = 1$ means that each target image is paired with one source image), which is 8579 (around 10%). These empirical results indicate that $H$ may provide a "soft" coupling relationship, since there may exist multiple source images with "non-zero $H$" for most target images.

**Stability and convergence of training process for learning $u_\omega$ and $v_\omega$.** We show the objective function (Eq. (6)) in training in Figs. A-2(a-b). We can see that the objective function first increases

and then converges, under learning rates 1e-5 and 1e-6. We notice that different $u_\omega$ and $v_\omega$ may yield the same $H$, (*e.g.*, $u_\omega(\mathbf{x}) + c$ and $v_\omega(\mathbf{y}) - c$ yield the same $H(\mathbf{x}, \mathbf{y})$ as $u_\omega(\mathbf{x})$ and $v_\omega(\mathbf{y})$, as in Eq. (7)). We then show the relative change of $H$ in training in Fig. 2(c). We can see that the relative difference of $H$ first decreases and fluctuates near to zero, which may be because the optimization is based on approximated gradients over mini-batch. The $\frac{1}{\epsilon}$ ($\epsilon = $ 1e-5 or 1e-7 in experiments) in Eq. (6) may yield large gradients. We then choose a small learning rate to stabilize the training.

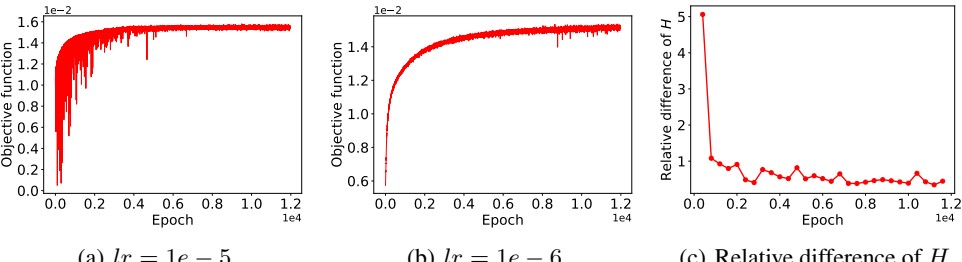

$$\text{(a) } lr = 1e - 5 \qquad \text{(b) } lr = 1e - 6 \qquad \text{(c) Relative difference of } H$$

Figure A-2: (a-b) Curves of objective function in Eq. (6) under learning rates $lr = 1e - 5$ and $lr = 1e - 6$ with $\epsilon = 1e - 5$. (c) Relative difference of $H$ in training. The relative difference of $H$ is defined as $\frac{\|H^i - H^{i+\Delta}\|_{p,q}}{\|H^i\|_{p,q}}$, where the norm $\|H\|_{p,q} = \left[\mathbb{E}_p \mathbb{E}_q H(\mathbf{x}, \mathbf{y})^2\right]^{1/2}$ (*i.e.*, the $L_2$-norm of functions on the sample space associated to measure $p \otimes q$). $H^i$ is the function at training step $i$. To reduce the computational cost, we set $\Delta = 10000$. $p$ and $q$ are distributions of source and target training data.

**On the choice of $\epsilon$.** To better approach the original OT in Eqs. (2-3) by the $L_2$-regularized OT in Eq. (5) so that the OT guidance could be better achieved, the $\epsilon$ should be small. However, due to the term $\frac{1}{\epsilon}$ in the objective function in Eq. (6), smaller $\epsilon$ may suffer from numerical issues in training. As a balance, we empirically choose a $\epsilon$ from candidate values 1e-5, 1e-6, 1e-7 such that the training is more stable. We show the objective function curves under varying $\epsilon$ in Fig. A-3. The training curves seem to be stable in general. We have also reported the results with varying $\epsilon$ in Table A-3. From Table A-3, we can see that FID ranges in [13.68, 14.56] (which seems to be stable) for $\epsilon$ in [1e-7,1e-3]. We can also see that Acc is similar for $\epsilon$ in 1e-7, 1e-6,1e-5, and decreases as $\epsilon$ increases from 1e-5 to 1e-3.

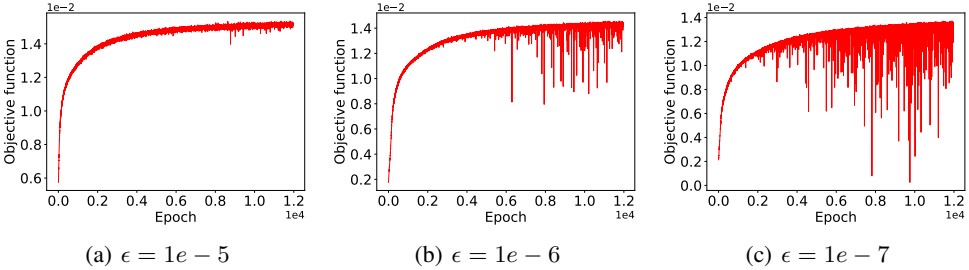

$$\text{(a) } \epsilon = 1e - 5 \qquad \text{(b) } \epsilon = 1e - 6 \qquad \text{(c) } \epsilon = 1e - 7$$

Figure A-3: The curves of objective function in training under varying $\epsilon$ with learning rate $lr = 1e-6$.

Table A-3: Results of OTCS using varying $\epsilon$.

| $\epsilon$ | 1e-7 | 1e-6 | 1e-5 | 1e-4 | 1e-3 |
|---|---|---|---|---|---|
| FID $\downarrow$ | 14.56 | 14.12 | 13.68 | 13.52 | 13.91 |
| Acc $\uparrow$ | 95.11 | 96.00 | 96.44 | 90.22 | 77.78 |

**Computational cost.** We report the computational time cost of our training process in this paragraph. As illustrated in Algorithm 2 in the Appendix A, our method consists of three processes in training:

(1) *learning the potentials* $u_\omega$ & $v_\omega$, (2) *computing $H$ & storing the target sample indexes with non-zero $H(H > 0.001)$ for each source sample*, and (3) *training the score-based model $s_\theta$*. We report the computational time cost of these three processes in the following Table A-4.

Table A-4: Computational time cost of training processes.

| Dataset | Learning $u_\omega$ & $v_\omega$ (30w steps) | Computing & storing $H$ | Training $s_\theta$ (60w steps) |
|---|---|---|---|
| CelebA | 3.5 hours | 0.5 hours | 5 days |
| Animal | 2.0 hours | 0.05 hours | 5 days |

From Table A-4, we can see that (1) learning $u_\omega$ & $v_\omega$ and (2) computing & storing $H$ takes no more than 4 hours. Similarly to the other diffusion approaches, (3) training our score-based model $s_\theta$ takes a few days.

***Computational time of each operation in a single step of training $s_\theta$.*** In each step of training the score-based model $s_\theta$, we sequentially (1) sampling the index of target sample with probability proportional to $H$ for a randomly selected source sample index (*sampling index*), then (2) load corresponding images (*loading image*), and (3) finally feed data to network and update model parameters (*updating network*). Compared with the training of score-based model for paired setting, our training additionally contains the operation of sampling index. From Table A-5, we can see that sampling index takes much less time than updating network.

Table A-5: Computational time of operations in a single step of training $s_\theta$ on Animal dataset.

| Sampling index | Loading image | Updating network |
|---|---|---|
| 0.0005 seconds | 0.01 seconds | 0.7 seconds |

## D.2    Additional Results on Toy Data Experiments

**Results of OTCS under varying $\epsilon$.** We show the results of OTCS under varying $\epsilon$ in Fig. A-4. We can see that the histogram of generated samples by OTCS fits the estimated conditional transport plan when $\epsilon$ is 0.01, 0.001, and 0.0001.

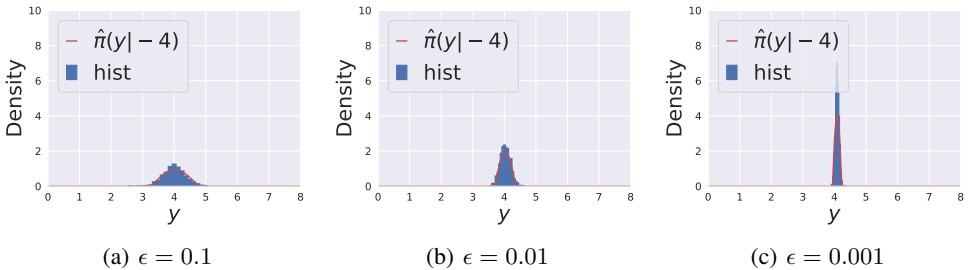

(a) $\epsilon = 0.1$        (b) $\epsilon = 0.01$        (c) $\epsilon = 0.001$

Figure A-4: The histogram ("hist") of generated samples by our proposed OTCS and the estimated conditional transport plan $\hat{\pi}(\mathbf{y}| - 4)$ under varying $\epsilon$.

**Results of OTCS under varying conditions.** We show the results of OTCS for varying condition $\mathbf{x}$ in Fig. A-5. We can see that the histogram of generated samples by OTCS fits the estimated conditional transport plan for $\mathbf{x} = -5, -4, -3$.

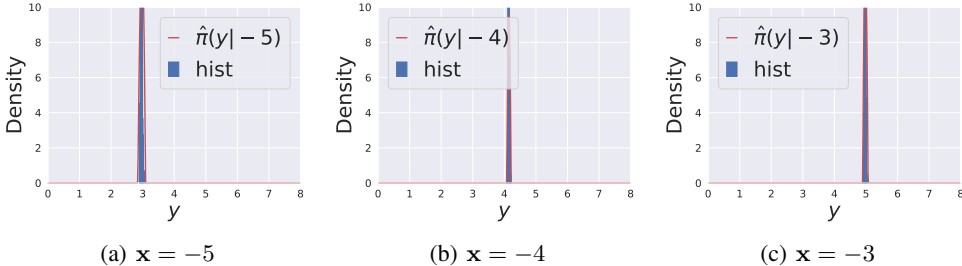

(a) $\mathbf{x} = -5$      (b) $\mathbf{x} = -4$      (c) $\mathbf{x} = -3$

Figure A-5: The histogram ("hist") of generated samples by our proposed OTCS and the estimated conditional transport plan $\hat{\pi}(\mathbf{y}|\mathbf{x})$ under varying condition $\mathbf{x}$. $\epsilon = 0.0001$ in this experiment.

### D.3    Additional Results in Unpaired Super-Resolution

**Results of different methods in unpaired super-resolution.** In Figs. A-6 and A-7, we visualize the translated images by our proposed OTCS, adversarial training-based OT methods of NOT and KNOT, and diffusion-based methods of SCONES, EGSDE, and DDIB. We can see that OTCS, NOT, and KNOT better preserve the identity/structure than SCONES, EGSDE, and DDIB. OTCS produces clearer translated images than NOT. The translated images by KNOT have artifacts (please zoom in on the figure to see the artifacts).

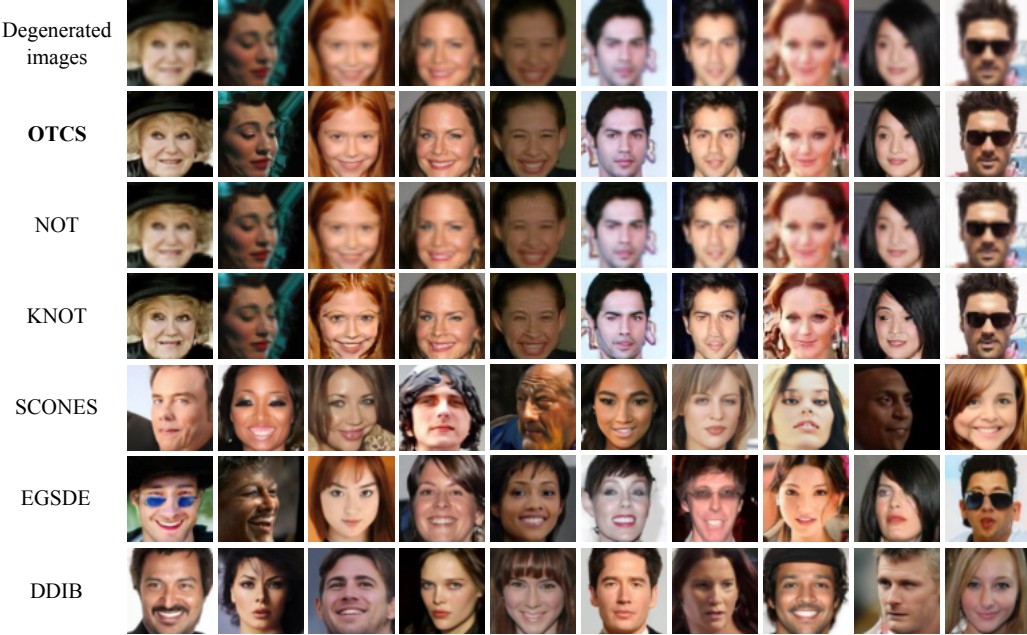

Figure A-6: Translated images by our proposed OTCS, adversarial training-based OT methods of NOT and KNOT, and diffusion-based methods of SCONES, EGSDE, and DDIB.

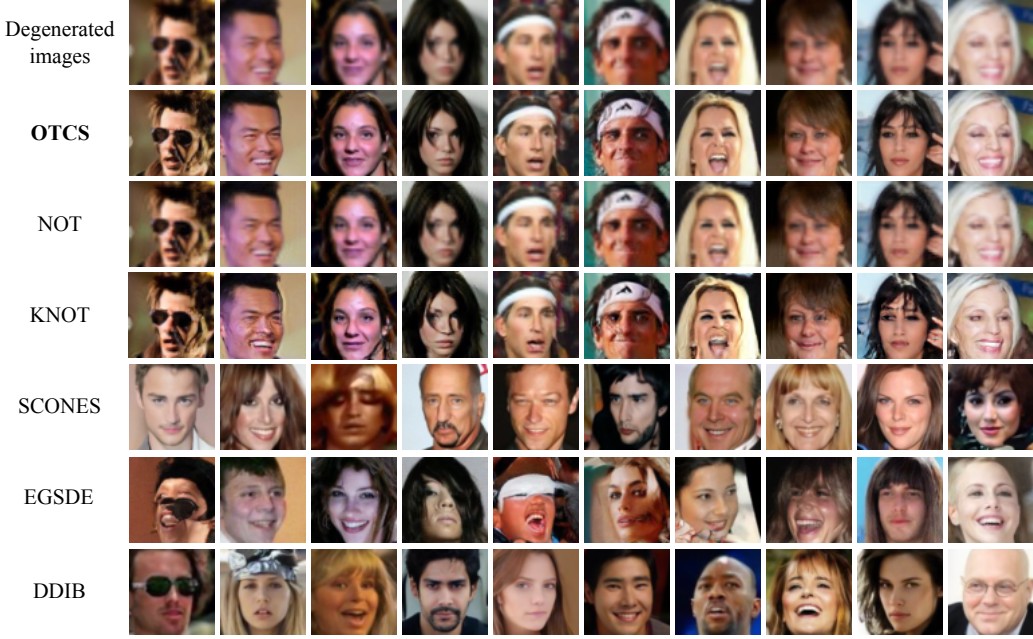

Figure A-7: Translated images by our proposed OTCS, adversarial training-based OT methods of NOT and KNOT, and diffusion-based methods of SCONES, EGSDE, and DDIB.

**Translated images by SCONES and OTCS with different initial strategies in reverse SDE in inference.** We show the translated images of SCONES and our proposed OTCS in Fig. A-8 with different initialization strategies in inference. We can observe that for the smaller initial noise scales (0.2 and 0.4), the translated images by SCONES are not realistic. For larger initial noise scales (0.8 and 1.0), the structures of translated images by SCONES are apparently different from those of degenerated images. The translated images by OTCS seem to be more realistic and share better structure similarity to degenerated images than SCONES, under different initial noise scales.

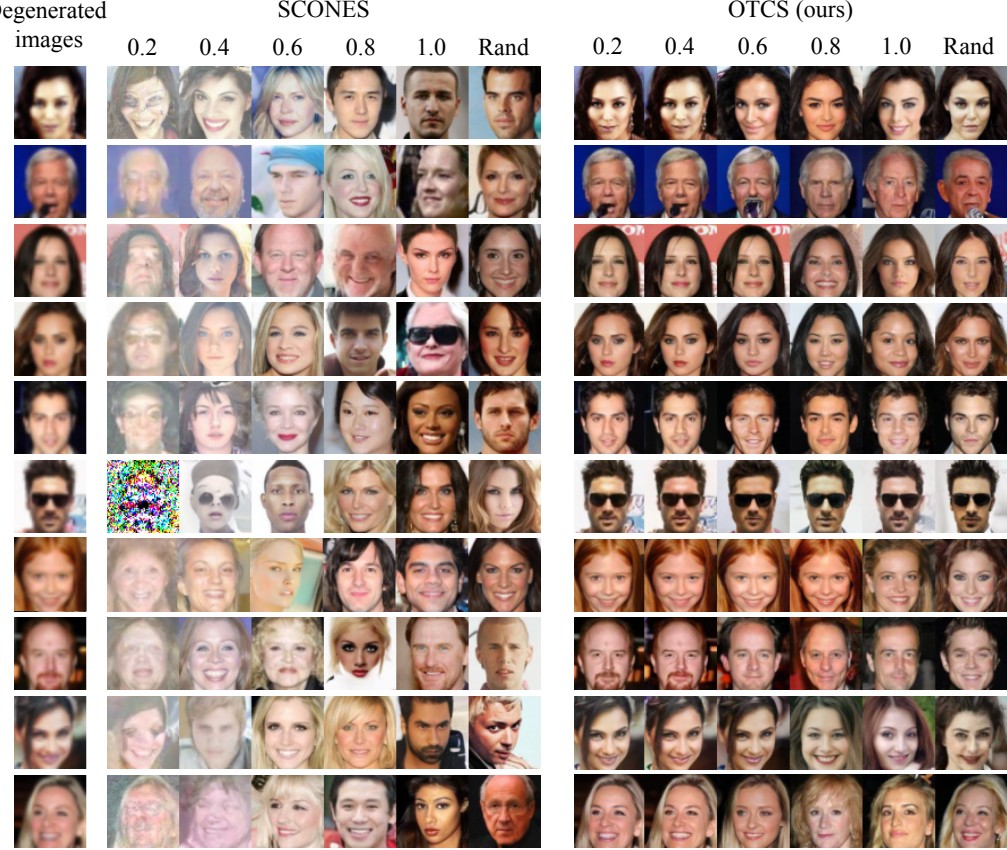

Figure A-8: Translated images by SCONES and OTCS with different initialization in inference. We consider the following initialization strategies: 1) We sample a noisy data $\mathbf{y}_M$ from $p_{M|0}(\mathbf{y}_M|\mathbf{x})$ as initial state, and the reverse SDE starts at time $M$. $\mathbf{x}$ is the degenerated image and $M$ is set to 0.2, 0.4, 0.8, and 1.0; 2) We directly generate a random noise $\mathbf{y}_T$ from $\mathcal{N}(0, \mathbf{I})$ as initial state (denoted as "Rand").

## D.4   Additional Results in Semi-paired I2I

**Translated animal images by different methods.**    We provide translated animal images by different methods in Figs. A-9, A-10, and A-11. We can see that OTCS better translates the source images to high-quality target images of desired species than the other methods.

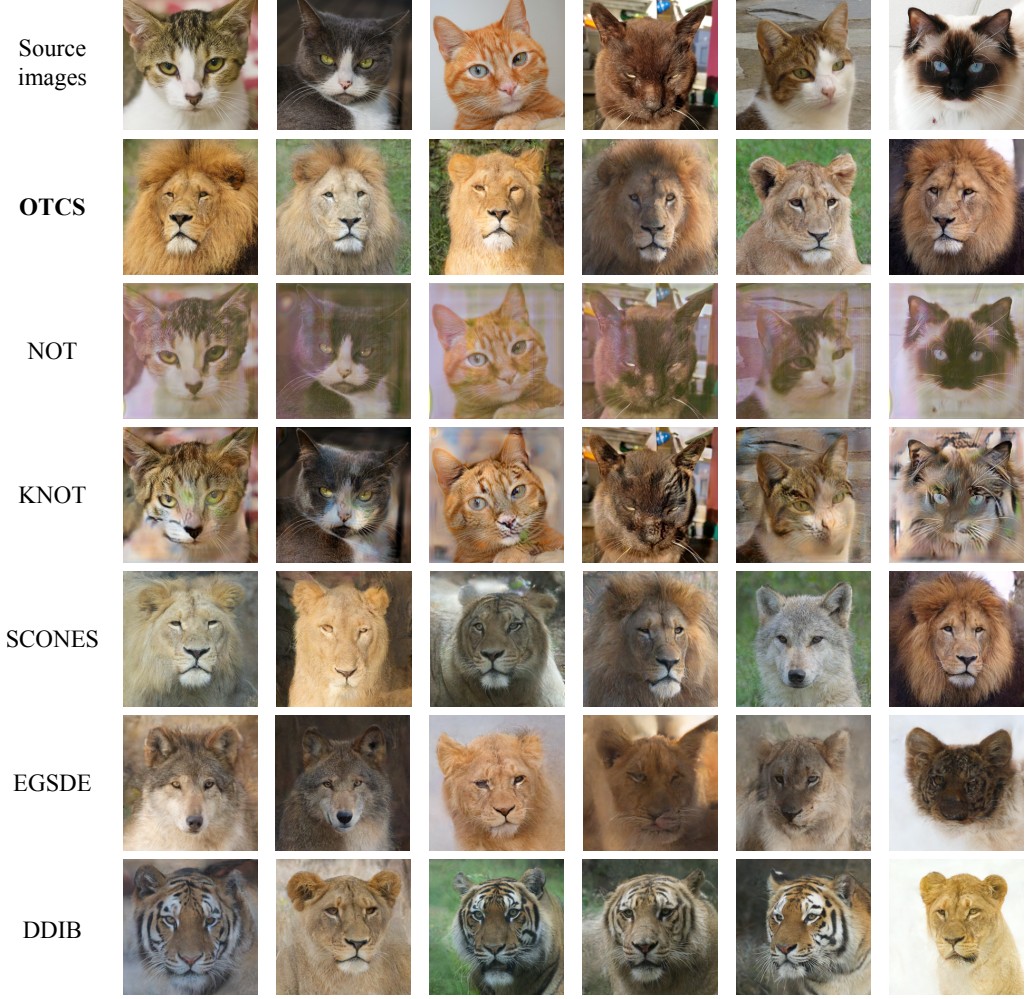

Figure A-9: Translated images of cat by different methods. With the guidance of paired images, we expect the images of cat to be translated into images of lion.

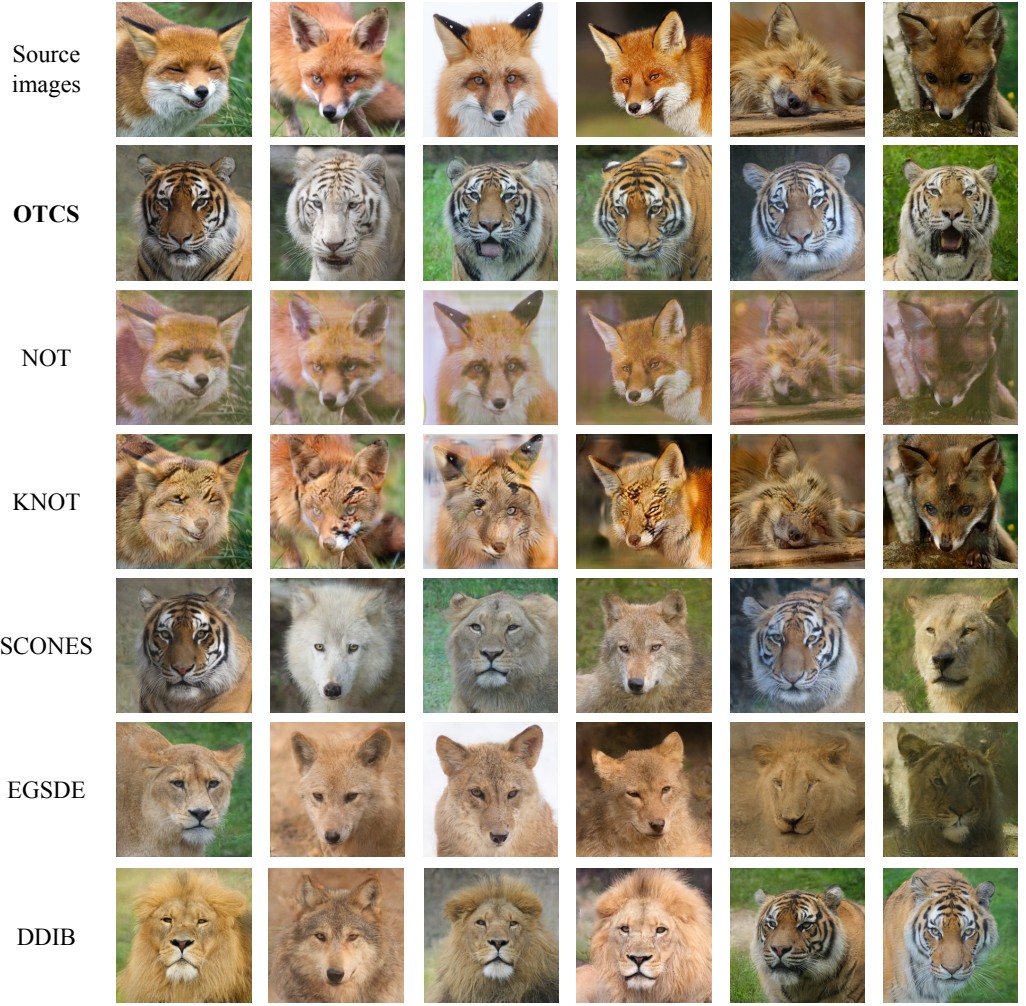

Figure A-10: Translated images of fox by different methods. With the guidance of paired images, we expect the images of fox to be translated into images of tiger.

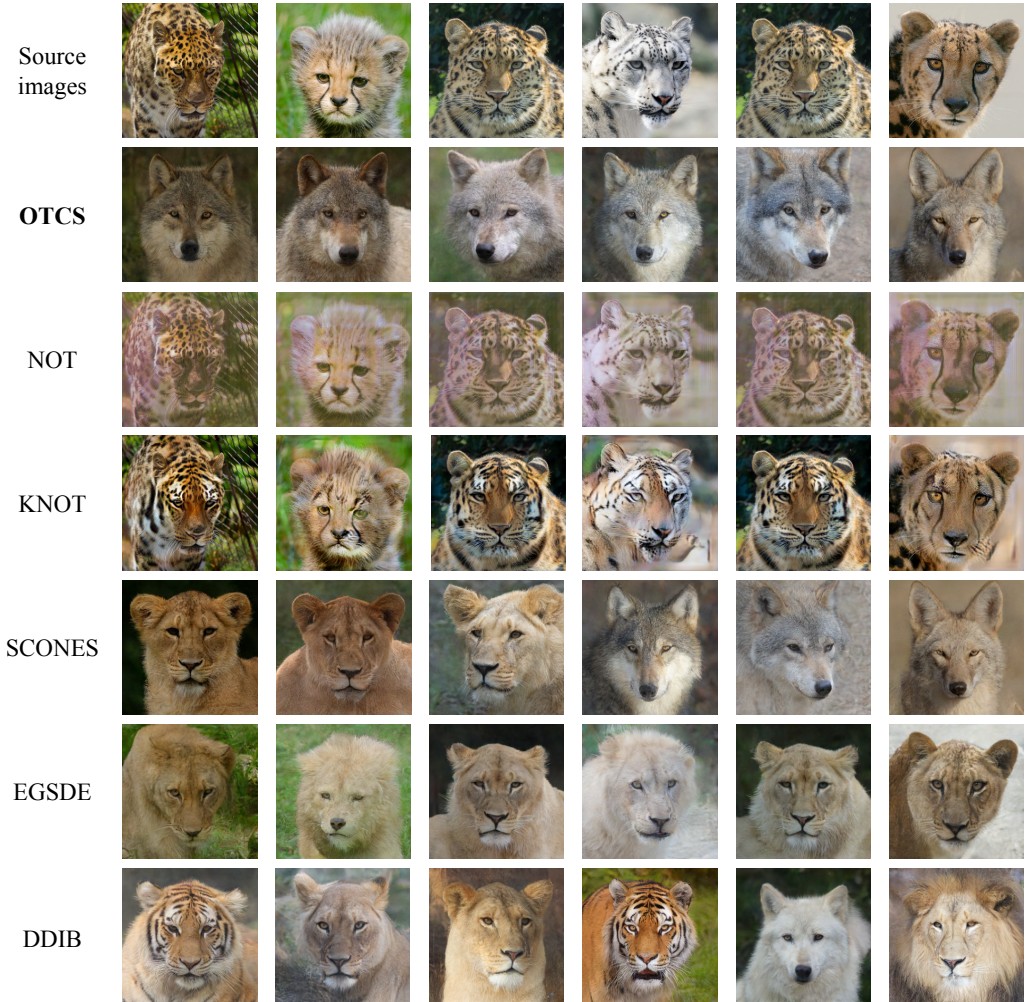

Figure A-11: Translated images of leopard by different methods. With the guidance of paired images, we expect the images of leopard to be translated into images of wolf.

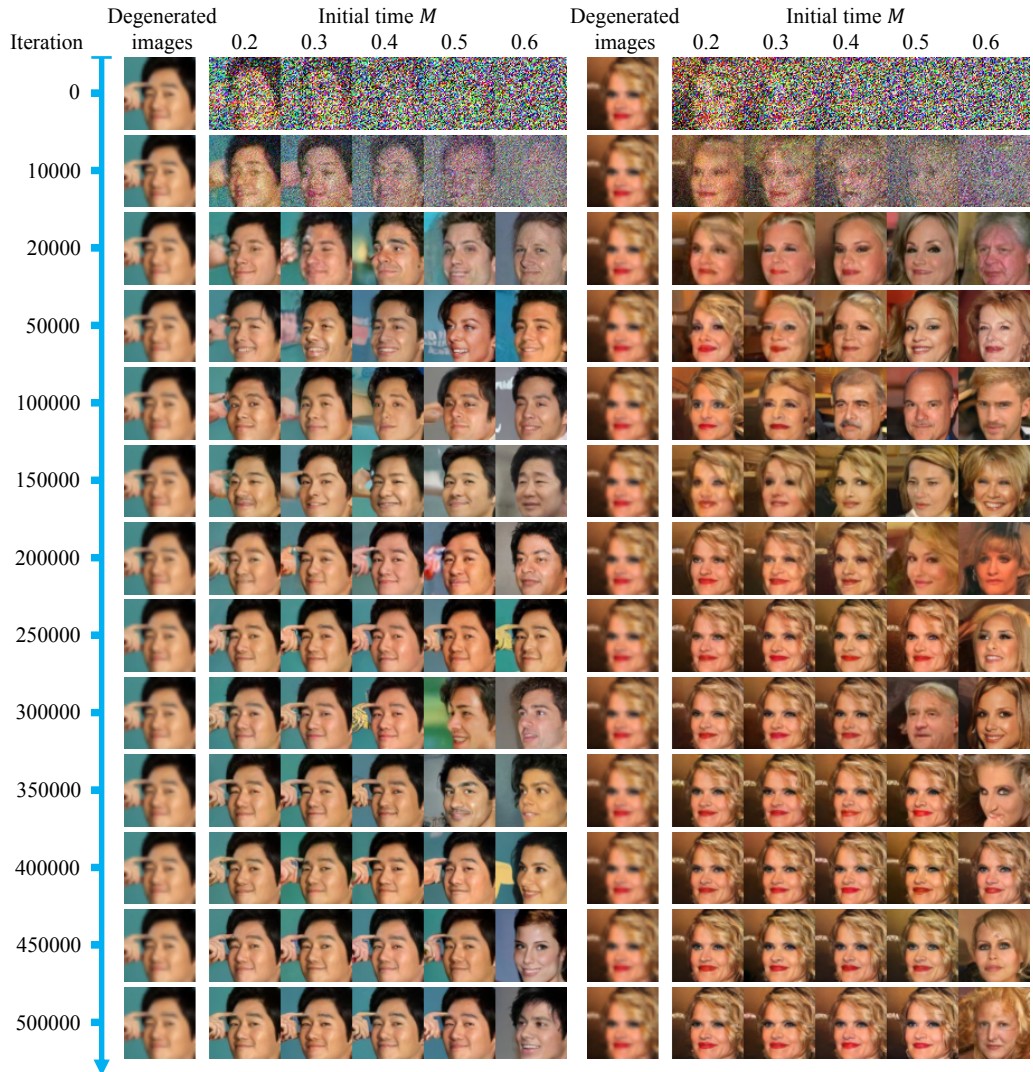

Figure A-12: Translated images by OTCS using models at varying training steps, in which we consider different initial time in reverse SDE for generating samples.

## D.5 Results for Trained Models at Different Training Steps

In Figs. A-12, A-13, and A-14, we show the translated images by OTCS in unpaired super-resolution using trained models at varying training steps, in which we consider different initial time $M$ in reverse SDE for generating samples.

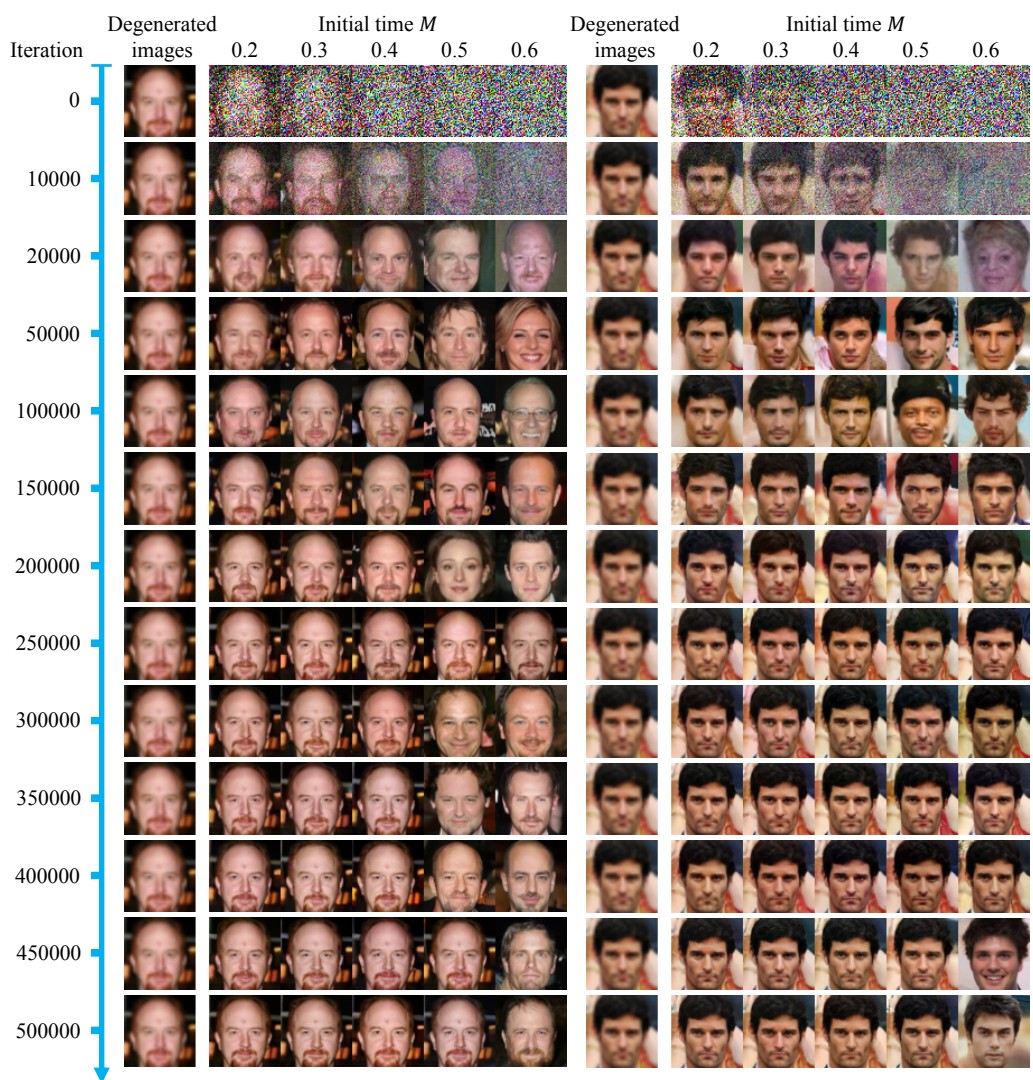

Figure A-13: Translated images by OTCS using trained models at varying training steps, in which we consider different initial time in reverse SDE for generating samples.

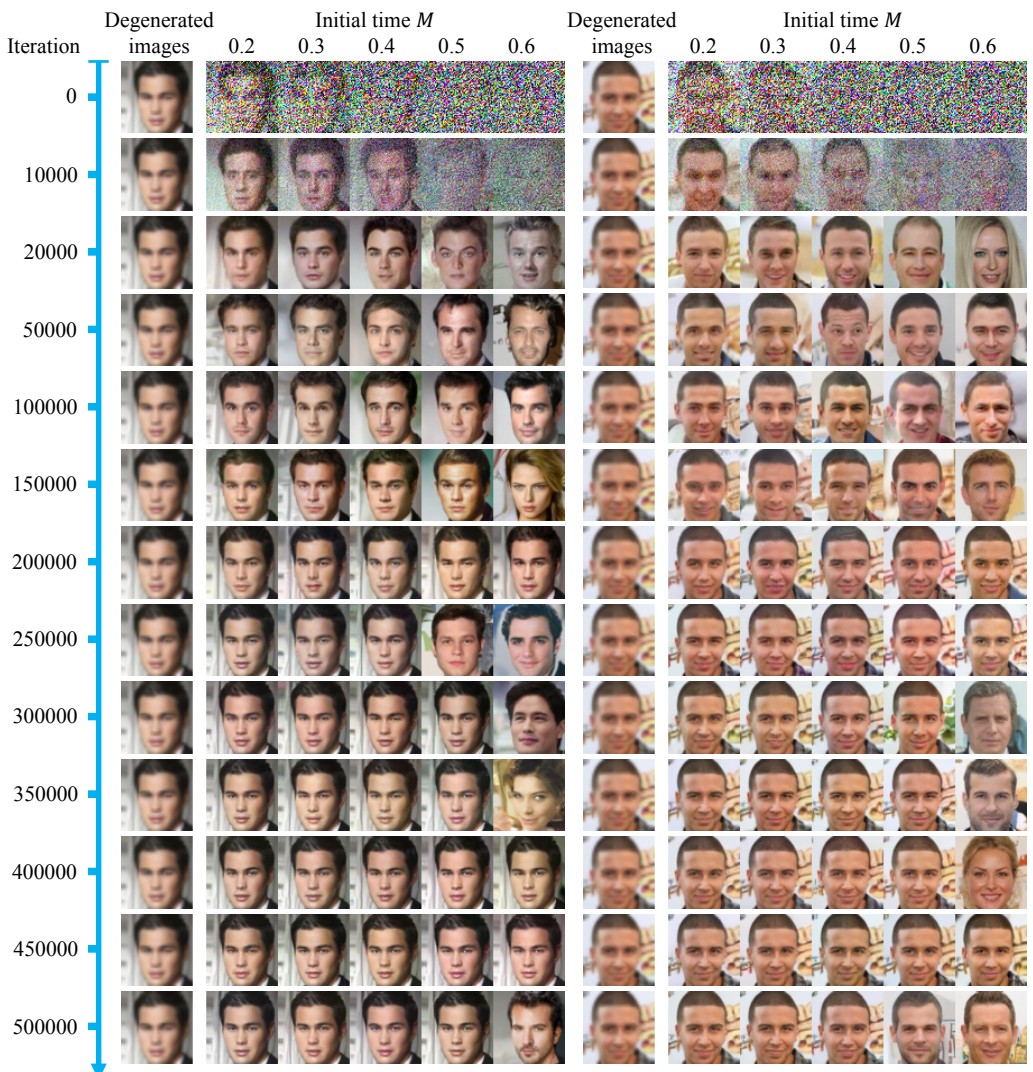

Figure A-14: Translated images by OTCS using models at varying training steps, in which we consider different initial time in reverse SDE for generating samples.