# OpenReview forum: "Optimal Transport-Guided Conditional Score-Based Diffusion Model"
_NeurIPS.cc/2023/Conference — NeurIPS 2023 poster_

### Official Review · Reviewer_Tt9x · 2023-06-11

**Soundness:** 4 excellent
**Presentation:** 3 good
**Contribution:** 4 excellent
**Rating:** 7
**Confidence:** 4

**Summary:**

This paper introduces a novel approach for training conditional score-based models. The proposed approach integrates the optimal-transport-based semi-supervised learning method into the popular score-based modeling framework with SDE sampling processes. The theoretical analysis presented in Section 4 showcases that the proposed method minimizes the 2-Wasserstein distance between the true and the modeled distributions. Furthermore, the experimental results in Section 5 demonstrate the superior performance of the proposed method in various applications, including super-resolution and image-to-image translation.

**Strengths:**

**Strengths**

In general, this paper is well-written, displaying a clear and logical structure. The definitions and formulations are presented with clarity and precision. The strengths of this paper are listed as follows:

1. The authors have provided a thorough discussion of the design choices made in the proposed framework, considering both practical and theoretical aspects. For example, in Section 3.3, they specifically address the potential drawback of directly implementing Eq. (9) and introduce the resampling-by-compatibility technique as a solution to this issue.

2. The experimental results as well as the ablation analyses provided in Section 5 demonstrate the effectiveness of the proposed method. The presented results showcase that the proposed OTCS approach achieves state-of-the-art performance across multiple applications, such as super-resolution and image-to-image translation.

3. There has been a growing interest in extending the conditional generation capabilities of score-based models to semi-supervised and unsupervised learning scenarios. This paper makes a significant contribution by providing a comprehensive exploration of this objective.


**Weaknesses:**

**Weaknesses**

However, the following aspects of the paper can be improved:

1. Clarifying how to calculate $H$ and its computational cost seems necessary (see Question 1).

2. The proposed method introduces a number of hyper-parameters. Specifying the approach of determining these values should be required (see Questions 2 and 3).

**Questions:**

**Questions**

1. The computation of $H(\mathbf{x}, \mathbf{y})$ plays a critical role in the proposed objective function and the resampling-by-compatibility technique. However, when dealing with a large value of $K$, it can become computationally intensive due to the calculation of the Jensen-Shannon (JS) divergence between two densities specifies in Eq. (4). Could the authors provide the detailed information regarding the computational cost of their training method?

2. It seems like that the optimization process of Eq. (6) may be unstable as the objective maximizes both $u_{\omega}(\mathbf{x})$ and $v_{\omega}(\mathbf{y})$ during training. In addition, it is observed that the learning rates are set to very small values (e.g., 1e-5 and 1e-6) throughout the experiments. Could the authors explain why the learning rates are set to such extremely small values? Does $\omega$ stably converge to its optimal value?

3. The selection of the regularization factor $\epsilon$ seems to be a crucial aspect in Eqs. (5) and (6). Could the authors elaborate on how to determine the regularization factor $\epsilon$? Is the training stability and evaluation accuracies of $u_{\omega}, v_{\omega}$ sensitive to the value of this hyper-parameter?

4. The experimental results reveal a significant performance difference between SCONES and OTCS. To further understand the factors contributing to this discrepancy, it would be helpful to clarify whether the model architecture used in SCONES is the same as that adopted in OTCS.


**Limitations:**

**Suggestion**

The illustration in Figure 1 lacks some crucial information. To improve clarity and provide a more comprehensive understanding of the proposed approach, the reviewer recommends including the 2-stage training procedure with their respective objective functions in the figure. Additionally, depicting the sampling process, including the SDE for conditional generation, would further enhance the visual representation.

**Minor Error**

There is a typo in line 108 (i.e., $R^s_\mathbf{y} \to R^t_\mathbf{y}$).

---

> ### Author Rebuttal · Authors · 2023-08-09
>
> We thank the reviewer for the insightful comments and suggestions on our paper.
>
> **Q1: Computational cost of our training method.**
>
> In our experiments, considering that the number of paired data could be small in real-world applications, we set $K$ to 3 for animal images and 10 for digits. We will report the computational time cost of our training process below. Note that for large $K$, we may calculate the relation in Eq. (4) using mini-batches of the paired data when training potentials $u_{\omega}$ and $v_{\omega}$.
>
> As illustrated in Algorithm 2 in the Appendix A.2, our method consists of three processes in training：(1) *learning the potentials $u_{\omega}$ & $v_{\omega}$*, (2) *computing $H$ & storing the target sample indexes with non-zero $H (H>0.001)$ for each source sample*, and (3) *training the score-based model $s_{\theta}$*. We report the computational time cost of these three processes in the following Table r4-1.
>
> Table r4-1: computational time cost of training processes.
>
> |Dataset|Learning $u_{\omega}$ & $v_{\omega}$ (30w steps)|Computing & storing $H$|Training $s_{\theta}$ (60w steps)|
> |:-:|:-:|:-:|:-:|
> |CelebA|3.5 hours|0.5 hours|5 days|
> |Animal|2.0 hours|0.05 hours|5 days|
>
> From Table r4-1, we can see that (1) learning $u_{\omega}$ & $v_{\omega}$ and (2) computing & storing $H$ takes no more than 4 hours. Similarly to the other diffusion approaches, (3) training our score-based model  $s_{\theta}$ takes a few days.
>
> **Computational time of each operation in a single step of training $s_{\theta}$**. In each step of training the score-based model $s_{\theta}$, we sequentially (1) sampling the index of target sample with probability proportional to $H$ for a randomly selected source sample index *(sampling index)*, then (2) load corresponding images *(loading image)*, and (3) finally feed data to network and update model parameters *(updating network)*. Compared with the training of score-based model for paired setting, our training additionally contains the operation of sampling index . From Table r4-2, we can see that sampling index takes much less time than updating network.
>
> Table r4-2: computational time of operations in a single step of training $s_{\theta}$ on Animal dataset.
>
> |Sampling index|Loading image|Updating network|
> |:-:|:-:|:-:|
> |0.0005 seconds|0.01 seconds|0.7 seconds|
>
> We will include the computational time cost in Appendix D, and cite them in Sect. 5.3.
>
>
>
> **Q2: Why the learning rates are set to such extremely small values? Does $\omega$ stably converge to its optimal value?**
>
> We show the objective function (Eq. (6)) in training in **Figs. r-1(a-b) of the uploaded pdf in the "global response"**. We can see that the objective function first increases and then converges, under learning rates 1e-5 and 1e-6. We notice that different $u_{\omega}$ and $v_{\omega}$ may yield the same $H$, (e.g., $u_{\omega}(\mathbf{x})+c$ & $v_{\omega}(\mathbf{y})-c$ yield the same $H(\mathbf{x},\mathbf{y})$ as $u_{\omega}(\mathbf{x})$ & $v_{\omega}(\mathbf{y})$, as in Eq. (7)). We then show the relative change of $H$ in training in Fig. r-1(c\). We can see that the relative difference of $H$ first decreases and fluctuates near to zero, which may be because the optimization is based on approximated gradients over mini-batch. The $\frac{1}{\epsilon}$ ($\epsilon=$ 1e-5 or 1e-7 in experiments) in Eq. (6) may yield large gradients. We then choose a small learning rate to stabilize the training. These results will be included in Appendix D, and cited in Sect. 5.3.
>
> **Q3: Could the authors elaborate on how to determine the regularization factor $\epsilon$? Is the training stability and evaluation accuracies of $u_{\omega},v_{\omega}$ sensitive to the value of this hyper-parameter?**
>
> To better approach the original OT in Eqs. (2-3) by the $L_2$-regularized OT in Eq. (5) so that the OT guidance could be better achieved, the $\epsilon$ should be small. However, due to the term $\frac{1}{\epsilon}$ in the objective function in Eq. (6), smaller $\epsilon$ may suffer from numerical issues in training. As a balance, we empirically choose a $\epsilon$ from candidate values {1e-5, 1e-6, 1e-7} such that the training is more stable.  We show the objective function curves under varying $\epsilon$ in **Fig. r-2 in the uploaded pdf**. The training curves seem to be stable in general. We have also reported the results with varying $\epsilon$ in Table A-1 in Appendix D.3. From Table A-1, we can see that FID ranges in [13.68, 14.56] (which seems to be stable) for $\epsilon$ in [1e-7,1e-3]. The Acc (reflecting how well the OT guidance is imposed) is {95.11,96.00,96.44,90.22,77.78} for $\epsilon$ in {1e-7,1e-6,1e-5,1e-4,1e-3}.  We can see that Acc is similar for $\epsilon$ in {1e-7, 1e-6,1e-5},  and decreases as $\epsilon$ increases from 1e-5 to 1e-3. These results will be included in Appendix D, and cited in Sect. 5.3.
>
>
>
> **Q4: Clarifying whether the model architecture used in SCONES is the same as that adopted in OTCS.**
>
> Thanks for this suggestion. We clarify that for fair comparisons, SCONES and OTCS utilize the same architecture of the score-based model (excepting that OTCS takes an additional block for conditioning on source data), the same trained potentials $u_{\hat{\omega}}$ & $v_{\hat{\omega}}$. This clarification will be included in the last paragraph in Sect. 4.
>
> **Q5: Revision of Fig. 1.**
>
> Thanks for this suggestion. As suggested, we will add two dotted frames in Fig. 1 for splitting the workflow into two stages, of which the titles are respectively "Stage I: $\min_{\omega} \mathcal{F}\_{\rm OT}(u_{\omega},v_{\omega})$" and "Stage II: $\min_{\theta}\mathcal{J}\_{\rm CDSM}(\theta)$". We use $\mathcal{F}_{\rm OT}(u,v)$ to denote the objective function of $L_2$-regularized OT in Eq. (6) for convenience. Meanwhile, we will add the equations of forward and reverse SDEs to Fig. 1.
>
> **Q6: About the typo.**
>
> Thanks for this suggestion. We will correct it in the revised paper.

---

> > ### Comment · Reviewer_Tt9x · 2023-08-11
> > **Response to the Authors**
> >
> > I appreciate the comprehensive response provided by the authors, which fully addressed my questions.

---

> > > ### Author Response · Authors · 2023-08-11
> > > **Thanks to Reviewer Tt9x**
> > >
> > > Thanks again for the valuable review, enabling us to enhance our paper.

---

### Official Review · Reviewer_63Ma · 2023-06-24

**Soundness:** 4 excellent
**Presentation:** 3 good
**Contribution:** 3 good
**Rating:** 8
**Confidence:** 4

**Summary:**

This paper studied the conditional score-based diffusion model (SBDM), which is an important topic in current machine learning research, and has great potential for real-world application. To overcome the limits in existing methodology, i.e., requirements of paired data and lacks methodology for conditional model, this paper proposes an optimal transport (OT) guided diffusion model, where the transport plan/coupling serves as: 1) a reasonable proxy for the conditional relation when paired data are inaccessible; 2) guidance and information for implementing the conditional score-based diffusion process. Theoretical results are derived to provide an intuitive understanding of the proposed method, and comprehensive experiments are conducted to validate the empirical efficiency.

**Strengths:**

+ The paper is well-written and easy to follow, where the problems, motivations, and solutions are clearly stated.
+ An OT-guided diffusion model is proposed, which addresses the main challenges in existing research.
+ Theoretical results guarantee the interpretability, learnability, and bounded property of the OT-guided model.
+ Significant empirical results in I2I translation application.


**Weaknesses:**

**1.** The justifications for OT-guided model could be enhanced. In Eq. (9) and Section 3.2, the OT-guided model is defined, where the difference between $\mathcal{J} _{DSM}$ and $\mathcal{J} _{CDSM}$ is the “weights” $H$ and $\mathcal{C}$. The authors claim that $H$ and $\mathcal{C}$ can be considered as soft and hard relationships, while some further discussions are highly expected, e.g., is the model with soft weight $H$ still a diffusion model?

  **2.** Based on the question above, can the hard and soft relationships be analyzed under empirical scenarios? For example, does $H$ or the induced coupling $\pi$ have sparse (hard) or dense (soft) property?

  **3.** The $a_+$ operation in $L_2$-regularized (i.e., Eq. (6)) seems to be redundant. Since there is already a square operation in $u(x)+v(y)-\xi (x,y)$, the values of function are strictly non-negative.

  **4.** About clarity. (1) Abbreviations are used before giving definition, e.g., SDE. (2) The main body and appendix are generally independent; thus, some assumptions should be clarified in the main body, e.g., assumptions in Thm. 2.

  **5.** The limitations of the proposed method should be discussed, e.g., under what scenario will the OT fail to capture the relation information, and how the guidance of OT affects the model performance? Some brief discussion and conclusion are highly expected.

----
--**Post-Rebuttal Update**--
After checking the responses by authors, the concerns are addressed and overall quality is improved, including: clarity on writing and limitation (W4, W5), mathematical rigor on diffusion model and function property (W1, W3), in-depth analysis on the property of coupling under different space $H$. Considering the response and other reviews, I agree that this paper is technically solid and has certain merits. Thus, I would like to raise the score and hope the authors will carefully incorporate the reviewers' suggestions and their responses.



**Questions:**

I would like the authors to address the concern in **weaknesses**.

**Limitations:**

The limitations and broader impacts are not discussed.

---

> ### Author Rebuttal · Authors · 2023-08-08
>
>
> We thank the reviewer for the insightful comments and suggestions on our paper.
>
> **Q1: The justifications for OT-guided model could be enhanced. In Eq. (9) and Section 3.2, the OT-guided model is defined, where the difference between $J_{DSM}$ and $J_{CDSM}$ is the “weights” $\mathcal{C}$ and $H$. The authors claim that $H$ and $\mathcal{C}$ can be considered as soft and hard relationships, while some further discussions are highly expected, e.g., is the model with soft weight still a diffusion model?**
>
> We replace $\mathcal{C}$ in $J_{ DSM}$ in Eq. (8) by $H$ to develop the formulation $J_{CDSM}$ in Eq. (9) to train the conditional score-based model $s_{\theta}$ for unpaired or partially paired settings. We next analyze that such an OT-guided model is still a diffusion model. Specifically, as we show theorem 1 in Sect. 3.5, $J_{CDSM}$ shares the same gradient w.r.t. $\theta$ with $J\_{CSM}$ defined in theorem 1. $J_{CSM}$ corresponds to the following diffusion model. Given a condition data $\mathbf{x}$, we take the learned conditional transport plan $\hat{\pi}(\mathbf{y}|\mathbf{x})=H(\mathbf{x},\mathbf{y})q(\mathbf{y})$ as initial distribution $p_0(\mathbf{y}_0|\mathbf{x})$ to generate a target sample $\mathbf{y}_0$. We then perform the forward SDE $d\mathbf{y}=f(\mathbf{y},t)dt+g(t)d\mathbf{w}$ to obtain noisy sample $\mathbf{y}$ at time $t$, and train the conditional score-based model $s\_{\theta}(\mathbf{y};\mathbf{x},t)$  to approximate $\nabla\log p_t(\mathbf{y}|\mathbf{x})$ by minimizing $J\_{CSM}$ (or equivalently $J\_{CDSM}$).
> The corresponding reverse SDE is $d\mathbf{y}=[f(\mathbf{y},t)-g(t)^2 \nabla\log p_t(\mathbf{y}|\mathbf{x})]dt+g(t)d\mathbf{w}$$\approx [f(\mathbf{y},t)-g(t)^2 s\_{\theta}(\mathbf{y};\mathbf{x},t)]dt+g(t)d\mathbf{w}$, which is consistent with our inference process in Eq. (10). We will summarize these discussions and include them in Sects. 3.2 and 3.5.
>
> **Q2: Based on the question above, can the hard and soft relationships be analyzed under empirical scenarios? For example,does $H$ or the induced coupling $\pi$ have sparse (hard) or dense (soft) property?**
>
> To study how sparse $H$ is, for each target image $\mathbf{y}$, we denote the number of source image $\mathbf{x}$ with "non-zero $H$" as $n_{\mathbf{y}}$ (i.e., $n_{\mathbf{y}}=|\\{\mathbf{x}:H(\mathbf{x},\mathbf{y})>0.001\\}|$, considering numerical issues) in CelebA dataset. The histogram of $n_{\mathbf{y}}$ is shown in the Table r3-1.
>
> Table r3-1: histogram of number $n_{\mathbf{y}}$ of source images with "non-zero $H$" for target image $\mathbf{y}$, where the total numbers of both source images and target images are 80k.
>
> |Bins for $n_{\mathbf{y}}$|[0,10)|[10,20)|[20,50)|[50,100)|[100,600)|[600,80k]|
> |:-:|:-:|:-:|:-:|:-:|:-:|:-:|
> |Frequency|59600|8064|8468|3537|1716|0|
>
> We can see from Table r3-1 that all the target images are with $n_{\mathbf{y}}\leq 600$, and more than 70% of target images are with  $n_{\mathbf{y}}\leq 10$. This implies that for each $\mathbf{y}$ in more than 70% target images, there are no more than 10 among 80K source images $\mathbf{x}$ satisfying $H(\mathbf{x},\mathbf{y})>0.001$. So $H$ is sparse to some extent. We also count the number of target images with $n_{\mathbf{y}}=1$ ($n_{\mathbf{y}}=1$ means that each target image is paired with one source image), which is 8579 (around 10%). These empirical results indicate that $H$ may provide a ''soft" coupling relationship, since there may exist multiple source images with "non-zero $H$" for most target images.  We will include this empirical analysis in Appendix D (Table r3-1 will be displayed as a bar plot), and cite it in Sect. 3.2 of the revised paper.
>
> **Q3: The $a_+$operation in $L_2$-regularized OT (i.e., Eq. (6)) seems to be redundant. Since there is already a square operation in $u(\mathbf{x})+v(\mathbf{y})-\xi(\mathbf{x},\mathbf{y})$, the values of function are strictly non-negative.**
>
> Sorry for this misunderstanding. In Eq. (6), the symbol $(\cdot)\_+^2$ means that we first perform the $a_+$ operation and then the square operation. We will use $[(\cdot)_+]^2$ in the revised paper.
>
>
>
> **Q4: About clarity. (1) Abbreviations are used before giving definition, e.g., SDE. (2) The main body and appendix are generally independent; thus, some assumptions should be clarified in the main body, e.g., assumptions in Thm. 2.**
>
> Thanks for the suggestions.
>
> (1) We will use "stochastic differential equation (SDE)" to replace "SDE" Sect. 2.1 where "SDE" is used for the first time, in the revised paper.
>
> (2) In Appendix B.3.1, assumptions (1)-(8) are based on assumptions in Kwon et al. (NeurIPS,2022), most of which are used to ensure the existence of solution of SDEs. Assumption (9) states that the Lagrange function is $\kappa$-strongly convex in $L_1$-norm w.r.t. $\pi$, which is true for some $\kappa$ as shown in Daniels et al. (NeurIPS, 2021). As suggested, we will include assumption (9) in the second paragraph in Sect. 4 in the revised paper. Meanwhile, we will cite Kwon et al. (NeurIPS, 2022) and mention assumptions (1)-(8) in this paragraph.
>
>
>
> **Q5: Some brief discussion and conclusion on limitations are expected.**
>
> Thanks for the suggestion. When applying our approach, the cost function should be determined first. In our experiments, we simply choose the squared $L_2$-distance in image space for unpaired super-resolution and in feature space for semi-paired I2I (please refer to Q2 for reviewer H2CC for the details), and achieved satisfactory performance. However, if more domain knowledge is employed to define the cost function, the performance may be improved.  Meanwhile, if the number of target data is small, the generation ability of our trained model may be limited.  As suggested, we will include the discussion on limitations in the revised paper following the conclusion.

---

> > ### Comment · Reviewer_63Ma · 2023-08-13
> > **Concerns are addressed.**
> >
> > I thank the authors for the detailed justifications, which improve the clarity, mathematical rigor, and empirical validation. After checking other reviews, I think this paper is technically solid and the responses have addressed my concerns. Thus, I would like to raise the score and hope the authors will carefully incorporate the reviewers' suggestions and their responses.

---

> > > ### Author Response · Authors · 2023-08-14
> > > **Thanks to Reviewer 63Ma**
> > >
> > > We thank the reviewer again for the valuable review, which enables us to enhance the paper. As suggested, we will carefully revise the paper according to the suggestions of reviewers and the responses.

---

### Official Review · Reviewer_w9nv · 2023-07-06

**Soundness:** 3 good
**Presentation:** 3 good
**Contribution:** 3 good
**Rating:** 7
**Confidence:** 4

**Summary:**

This paper propose Optimal Transport-guided Conditional Score-based diffusion model (OTCS), which is a novel approach in training unpaired / semi-paired image-to-image translation network. The proposed approach leverage unsupervised / semi-supervised optimal transport to earn coupling between images and use such information to guide during training. The proposed OT-guided Conditional Denoting Score Matching objective recovers original Conditional Denoising Score Matching objective if the transport plan is accurate. Through experiments on unpaired or semi-paired image-to-image translation tasks, the proposed OTCS achieves state-of-the-art result.

**Strengths:**

The proposed approach is clearly described and present a novel idea to overcome the requirement of paired images in training image-to-image translation networks. The author successfully fuse optimal transport into Denoising Score Matching objective with theoretical guarantee. Also, to dodge the inexistent of target samples in batch training, the author propose Resampling-by-compatibility, which is a novel component of OTCS that aims to select good target sample during training. Overall, the paper is written clearly and the experimental results strongly demonstrate the effectiveness of the proposed method.

**Weaknesses:**

Since the method requires estimation of compatibility function to solve optimal transport, it requires additional computational cost (e.g., learning u and v in solving dual problem). Also, the resampling-by-compatibility include additional sampling stage. Lastly, the experimental setup is somewhat synthetic that there always exist good target in the dataset. It would be nice if there is more real-world unpaired / semi-paired image-to-image translation task.

**Questions:**

Questions
1. What if there is no good target for some source data? Then is the compatibility function becomes zero? In other words, how good is the compatibility function is calibrated?
2. I think the proposed approach can be generalized into training of text-to-image score-based models. For example, given a bunch of image-caption pairs and unpaired image-only dataset, the proposed OTCS can be used to further improve the generative quality of diffusion models. Do the authors have ever tried to do such kind of task?


**Limitations:**

Explicitly confirming the limitation of this work can enrich the paper. For example, what should be considered when considering unpaired / semi-paired image-to-image translation task on wild or when do they fail (e.g., when target dataset is not sufficient).

---

> ### Author Rebuttal · Authors · 2023-08-08
>
> We thank the reviewer for the insightful comments and suggestions on our paper.
>
> **Q1: Computational cost of the training method.**
>
> As illustrated in Algorithm 2 in the Appendix A.2, our method consists of three processes in training：(1) *learning the potentials $u_{\omega}$ & $v_{\omega}$*, (2) *computing $H$ & storing the target sample indexes with non-zero $H (H>0.001)$ for each source sample*, and (3) *training the score-based model $s_{\theta}$*. We report the computational time cost of these three processes in the following Table r2-1.
>
> Table r2-1: computational time cost of training processes.
>
> |Dataset|Learning $u_{\omega}$ & $v_{\omega}$ (30w steps)|Computing & storing $H$|Training $s_{\theta}$ (60w steps)|
> |:-:|:-:|:--:|:--:|
> |CelebA|3.5 hours|0.5 hours|5 days|
> |Animal|2.0 hours|0.05 hours|5 days|
>
> From Table r2-1, we can see that (1) learning $u_{\omega}$ & $v_{\omega}$ and (2) computing & storing $H$ takes no more than 4 hours. Similar to the other diffusion approaches, (3) training our score-based model  $s_{\theta}$ takes a few days.
>
> **Computational time of each operation in a single step of training $s_{\theta}$**. In each step of training the score-based model $s_{\theta}$, we sequentially (1) sampling the index of target sample with probability proportional to $H$ for a randomly selected source sample index *(sampling index)*, then (2) load corresponding images *(loading image)*, and (3) finally feed data to network and update model parameters *(updating network)*. Compared with the training of conditional score-based model for paired setting, our training additionally contains the operation of sampling index. From Table r2-2, we can see that sampling index takes much less time than updating network.
>
> Table r2-2: computational time of operations in a single step of training $s_{\theta}$ on Animal dataset.
>
> | Sampling index| Loading image| Updating network|
> | :-: | :-: | :-: |
> | 0.0005 seconds|0.01 seconds|0.7 seconds|
>
> We will include the computational time cost in Appendix D, and cite them in Sect. 5.3.
>
> **Q2: The experimental setup is somewhat synthetic that there always exist good targets in the dataset. It would be nice if there is more real-world unpaired / semi-paired image-to-image translation tasks. What if there is no good target for some source data? Then is the compatibility function becomes zero? In other words, how good the compatibility function is calibrated?**
>
> We clarify that there does not exist corresponding paired target data for each source image in the training datasets, for all our experiments. For example, in the unpaired super-resolution on CelebA, the ground truth high-resolution image for each low-resolution image is not in the training dataset. To figure out what happens to the compatibility function $H$ when there is no good target data, we conduct the following experiments. Firstly, we count the number of source samples satisfying that there is no target sample such that $H>0.001$, in CelebA dataset. We find that **25.9%** of source samples meets such condition. Note that the other source samples are often with $H$ larger than 1000 on some target samples (the large $H$ may be because $\epsilon$ is 1e-7 in Eq. (7)). Secondly, we add noisy images to the source dataset to train the potentials $u_{\omega}$ and $v_{\omega}$, and count the ratio of noisy images satisfying $H<0.001$ on all target samples. The results are reported in Table r2-3. The noisy images are generated from the standard normal distribution and with the same shape as the source images.
>
> Table r2-3:  ratio of noisy images with $H<0.001$ when adding varying numbers of noisy images to the source dataset.
>
> |Number of noisy images : Number of clean images|0.1 : 1|0.2 : 1|0.3 : 1|0.4 : 1|0.5 : 1|
> |-|:-:|:-:|:-:|:-:|:-:|
> |**Ratio of noisy images assigned with $H<0.001$**|89.3%|85.6%|83.9%|81.6%|80.2%|
>
> It can be seen that more than 80% of noisy images that are with no good target data are assigned with near-to-zero $H$ ($H<0.001$), when the ratio of numbers of noisy images to clean images is in [0.1,0.5].
>
> In our experiments, we have used real-world images in both unpaired super-resolution and semi-paired I2I. In unpaired super-resolution, we follow the super-resolution methods [9] to downsample the high-resolution images to obtain low-resolution images. As suggested, we will investigate more real-world applications in our future work.
>
> We will include the above results and discussions in Appendix D and cite them in Sect. 5.3 in the revised paper.
>
> **Q3: Regarding to experiments on text-to-image translation.**
>
> Thanks for the suggestion. The text-to-image translation approaches (Saharia et al., NeurIPS, 2022; Rombach et al., CVPR, 2022) often embed the caption into the latent space, on which the score-based model is conditioned. These methods leverage large-scale paired text-image datasets, such as COCO, to train the score-based model, achieving satisfying performance (e.g., Stable Diffusion and Saharia et al., NeurIPS, 2022). We do believe it is interesting to investigate the performance of our approach when given only a small fraction of text-image pairs.  Since training the score-based models on large-scale text-image datasets is time-consuming for the limited time in the rebuttal, we leave it for further investigation.
>
> **Q4: Explicitly confirming limitations.**
>
> Thanks for the suggestion. The cost function should be determined first when using our method. In experiments, we simply choose the squared $L_2$-distance in image space for unpaired super-resolution and in feature space for semi-paired I2I, achieving satisfactory performance. However, if more domain knowledge is employed to define the cost function, the performance may be improved.  Meanwhile, if the number of target data is small, the generation ability of our trained model may be limited.  As suggested, we will include these limitations in the revised paper.

---

> > ### Comment · Reviewer_w9nv · 2023-08-17
> > **Response to the rebuttal**
> >
> > I sincerely appreciate the authors' effort on rebuttal despite the short period. I am satisfied with the authors' response to the questions and concerns that I mentioned, and I suggest acceptance.

---

> > > ### Author Response · Authors · 2023-08-17
> > > **Thanks to Reviewer w9nv**
> > >
> > > We thank the reviewer again for the valuable review. We will revise our paper as discussed in the response.

---

### Official Review · Reviewer_H2CC · 2023-07-07

**Soundness:** 3 good
**Presentation:** 3 good
**Contribution:** 3 good
**Rating:** 6
**Confidence:** 3

**Summary:**

This paper proposes to use the Optimal Transport to guide the training of the conditional score-based diffusion model. The method first computes the coupling of the source and target data and then includes the coupling into the conditional score-based model objective. CelebA face super-resolution, AFHQ animal, and MNIST digits translation experiments show the proposed method performs better than state-of-the-art methods.


**Strengths:**

1. The idea that adopts Optimal Transport for training Conditional Score-Based Diffusion Model (SBDM) seems novel. For unpaired or semi-paired source and target data, it is natural to use OT to find the couplings and then train the SBDM.

2. The authors reformulated the original conditional SBDM such that it can take the coupling between the source and target data points (Proposition 1). Based on the reformulation, the authors proposed the OT-guided conditional denoising score-matching objective in Eq. 9.

3. The authors analyzed that the proposed method approximately generates samples from the conditional transport plan and gave the upper bound of the expected Wasserstein distance between the distribution of generated samples and the optimal transport plan.

4. Unpaired Super-Resolution results on the CelebA dataset show that the method can restore degenerated face images. The FID and SSIM show that the method significantly outperforms other OT, Flow, and diffusion methods. Semi-paired I2I translation results show that the proposed method performs I2I well and outperforms other methods.


**Weaknesses:**

The OT part is computed first, and the couplings are fixed before training the diffusion model. So, the whole training pipeline is not end-to-end, which may yield sub-optimal results.

It's not clear what cost function, $c(\cdot, \cdot)$ in Eq. 4, is used to compute the source image and target image transport cost. Also, it's not clear where is the cost function applied. Is it applied in the image space?

From Fig. 4 (a), it is hard to tell why the translated animals are good. From my understanding, the method should generate the translated animal with the same pose as the animal in the source image.



**Questions:**

The authors use ”condition data“ as source data in I2I. In general, we refer to "source data" instead of condition data.

**Limitations:**

If the transport cost is applied to the image space, then the OT may not make much sense because the cost computed in the image space may not reflect the accurate semantic distance.

---

> ### Author Rebuttal · Authors · 2023-08-08
>
> We thank the reviewer for the insightful comments and suggestions on our paper.
>
> **Q1: The OT part is computed first, and the couplings are fixed before training the diffusion model. So, the whole training pipeline is not end-to-end, which may yield sub-optimal results.**
>
> In this paper, we aim to use OT to guide the learning of conditional score-based model for applications with unpaired or partially paired training dataset. To tackle the lack of coupling relationship for unpaired or partially paired settings, we use $L_2$-regularized OT to build the coupling relationship between source and target data, as mentioned by the reviewer in Strengths 1. The coupling relationship in $L_2$-regularized OT is modeled in the compatibility function $H(\mathbf{x},\mathbf{y})$ as shown in Eq. (7). To develop the formulation of conditional SBDM for unpaired or partially paired settings, as mentioned in the Strengths 2 by the reviewer, we provide a reformulation in Eq. (8) of conditional SBDM for paired setting, in which the coupling relationship of paired source/condition data $\mathbf{x}$ and target data $\mathbf{y}$ is explicitly modeled by function $\mathcal{C}(\mathbf{x},\mathbf{y})$.  This naturally motivates us to replace $\mathcal{C}(\mathbf{x},\mathbf{y})$ in Eq. (8) by $H(\mathbf{x},\mathbf{y})$ to extend this reformulation of paired setting (Eq. (8)) to unpaired or partially paired settings (Eq. (9)). Using Eq. (9) to train the conditional score-based model $s_{\theta}$, the guidance of OT is imposed. As demonstrated by the experimental results in Table 1, our approach is superior to the compared state-of-the-art approaches in unpaired super-resolution and semi-paired image-to-image translation experiments. Since the objective for learning compatibility function $H$ (i.e., by learning $u_{\omega},v_{\omega}$) and conditional score-based model $s_{\theta}$ is different, it is non-trivial to learn by end-to-end training.  We will strive toward this goal in the future.
>
> **Q2: It's not clear what cost function, $c(\cdot,\cdot)$ in Eq. (4), is used to compute the source image and target image transport cost. Also, it's not clear where is the cost function applied. Is it applied in the image space?**
>
> Sorry for missing the specification of cost function in the main body of paper (we have included it in Appendix C). For the unpaired super-resolution task in Sect. 5.1, we should preserve the local structure and image details of low-resolution images in the transported high-resolution counterparts. We then take the squared $L_2$-distance in image space as the cost function, as discussed in Appendix C.2. For the semi-paired image-to-image translation task in Sect. 5.2, we aim to transport the source image to the desired classes in the target domain, guided by the given cross-domain image pairs with desired paired class labels. To do this, as discussed in Appendix C.3, we map the images to the semantic feature spaces using a pre-trained feature extractor. The cost function is taken as the squared $L_2$-distance in the semantic feature space. For animal images, the feature extractor is taken as the CLIP image encoder. For digits, we train an auto-encoder on each domain and take the encoder part as the feature extractor. Due to space limits, we have included these details in Appendix C.  Since the paper and appendix are generally independent, we will clarify the specification of cost function in the first paragraphs in Sects. 5.1 and 5.2 in the revised paper.
>
> **Q3: From Fig. 4 (a), it is hard to tell why the translated animals are good. From my understanding, the method should generate the translated animal with the same pose as the animal in the source image.**
>
> We clarify that the setup of our considered semi-paired I2I is different from the setup of unpaired I2I [15-18,35,36]. Our goal is to transport the source images to the desired classes guided by a few paired images with desired paired class labels. While the unpaired I2I [15-18,35,36] aims to preserve the pose of source images when translated to target domain. As implied by the experimental results in Table 1, the recent unpaired I2I methods [17,35,36] and the semi-paired I2I method [14] struggle to translate source images to the desired classes. To employ our approach to tackle the semi-paired I2I, as discussed in Q2, we take the squared $L_2$-distance in the semantic feature space as the cost function. We use the image encoder of CLIP to extract the image features, which may mainly contain the class information of the image, because the CLIP features are good for classification as shown in the paper of CLIP. The experimental results in Table 1 and Fig. 4 show better performance achieved by our approach. It is interesting to investigate translating source images to desired classes and meanwhile preserving the poses. For this purpose, we may additionally utilize the features encoding pose information to compute the cost function. We leave this as our future work. We will summarize this clarification and include it in Sect. 5.3.
>
> **Q4: The authors use "condition data'' as source data in I2I. In general, we refer to "source data'' instead of condition data.**
>
> Thanks for this suggestion. As suggested, we will use "source data'' in the descriptions of I2I and experiments.

---

> > ### Comment · Reviewer_H2CC · 2023-08-14
> > **Reply to rebuttal**
> >
> > Thank the authors for the response! Most of my concerns are addressed. I also checked other reviews. The paper is technically solid and achieves good experimental results. The authors need to revise the paper as promised in the response.

---

> > > ### Author Response · Authors · 2023-08-14
> > > **Thanks to Reviewer H2CC**
> > >
> > > Thanks again for the valuable review. We will carefully revise the paper as promised in the responses.

---

### Author Rebuttal · Authors · 2023-08-09

Dear ACs and reviewers,

Thanks for handling our paper. We have responded to the comments of each reviewer individually and discussed how to revise our paper according to the comments and suggestions. Meanwhile, we attached a pdf file containing the figures as support material for the rebuttal.

Best wishes,

Authors

---

### Comment · Area_Chair_EDHf · 2023-08-11
**Author-Reviewer Discussion Period**

Thank you reviewers for your work in evaluating this submission, and thank you authors for responding to the reviewers’ questions and concerns. We have now entered the discussion period, which will run until August 16th.

In order to make the most of this discussion, I would encourage the reviewers to read the authors’ responses and the other reviews, to ask further clarification questions if they have them, and adjust their scores whenever relevant by the end of the discussion period. At the very least, please acknowledge that you have read the authors’ response to your review.

Thank you everyone for making the review process a fruitful, constructive, and civil process.

Your AC

---

### Decision · Program_Chairs · 2023-09-21

**Decision:**

Accept (poster)

**Comment:**

This paper proposed an optimal transport (OT)-based method for training conditional score-based diffusion models. The reviewers were impressed with the empirical results of the proposed method, particularly when compared to other OT, Flow, and diffusion methods. They expressed some concerns about the clarity of presentation and notation, sensitivity to hyperparameters, and some missing details about the formulation. The author's respose seems to have addressed most of these concerns.

I agree with the reviewers that the empirical results suggest this is a promising method. However, the paper should ackowledge that the connection between OT and diffusion models has been made before in prior work. For example, this connection is the main subject of **Understanding DDPM Latent Codes Through Optimal Transport** by Khrulkov et al 2022. This reference was unfortunately not raised by the reviewers, but should nevertheless be cited and discussed in this work. Despite this, a formal OT treatment of SCDM is still very much an open problem, towards which this submission makes some important steps, which ultimately makes me lean towards recommending acceptance.